# Endo-lysosomal assembly variations among human leukocyte antigen class I (HLA class I) allotypes

Eli Olson[1,2], Theadora Ceccarelli[1], Malini Raghavan[1,2]*

[1]Department of Microbiology and Immunology, Michigan Medicine, University of Michigan-Ann Arbor, Ann Arbor, United States; [2]Graduate Program in Immunology, Michigan Medicine, University of Michigan-Ann Arbor, Ann Arbor, United States

**Abstract** The extreme polymorphisms of human leukocyte antigen class I (HLA class I) proteins enable the presentation of diverse peptides to cytotoxic T lymphocytes. The canonical endoplasmic reticulum (ER) HLA class I assembly pathway enables presentation of cytosolic peptides, but effective intracellular surveillance requires multi-compartmental antigen sampling. Endo-lysosomes are generally sites of HLA class II assembly, but human monocytes and monocyte-derived dendritic cells (moDCs) also contain significant reserves of endo-lysosomal HLA class I molecules. We hypothesized variable influences of HLA class I polymorphisms upon outcomes of endo-lysosomal trafficking, as the stabilities and peptide occupancies of cell surface HLA class I molecules are variable. Consistent with this model, when the endo-lysosomal pH of moDCs is disrupted, HLA-B allotypes display varying propensities for reductions in surface expression, with HLA-B*08:01 or HLA-B*35:01 being among the most resistant or sensitive, respectively, among eight tested HLA-B allotypes. Perturbations of moDC endo-lysosomal pH result in accumulation of HLA-B*35:01 in LAMP1[+] compartments and increase HLA-B*35:01 peptide receptivity. These findings reveal the intersection of the vacuolar cross-presentation pathway with a constitutive assembly pathway for some HLA-B allotypes. Notably, cross-presentation of epitopes derived from two soluble antigens was also more efficient for B*35:01 compared to B*08:01, even when matched for T cell response sensitivity, and more affected by cathepsin inhibition. Thus, HLA class I polymorphisms dictate the degree of endo-lysosomal assembly, which can supplement ER assembly for constitutive HLA class I expression and increase the efficiency of cross-presentation.

*For correspondence: malinir@umich.edu

Competing interest: The authors declare that no competing interests exist.

## Editor's evaluation

This study provides novel insights into the role of HLA polymorphisms in the processing of exogenous antigens. The evidence supporting the conclusions is strong, with rigorous cellular and biochemical assays. The work will be of interest to scientists within the field of antigen presentation.

## Introduction

The major histocompatibility complex class I (MHC class I) molecule is responsible for providing nearly all cells in the body with the capability for intracellular immune surveillance. This is accomplished by presenting intracellular peptides on the cell surface, where they are recognized by CD8[+] T cells (*Bjorkman and Parham, 1990*). CD8[+] T cells that have been properly selected in the thymus do not react to self-derived peptides but may recognize and activate in response to foreign antigen (*Klein et al., 2014*). Assembly of MHC class I complexes with peptides typically occurs in the endoplasmic reticulum (ER), where nascent MHC class I heavy chains are synthesized and associated with

the invariant β$_2$m light chain. This is followed by their binding to a complex of chaperones known as the peptide loading complex (PLC), in close association with the transporter associated with antigen processing (TAP), a dimeric peptide transporter that brings cytosolically processed peptides into the ER for peptide loading. Upon successful assembly, MHC class I complexes are sufficiently stabilized so that they can release from the PLC and traffic to the cell surface (*Blum et al., 2013*; *Raghavan and Geng, 2015*). Assembly is a highly coordinated and regulated process that is often targeted by viruses and cancers to escape immune surveillance (*Hansen and Bouvier, 2009*; *Leone et al., 2013*; *Verweij et al., 2015*); thus highlighting the importance of efficient assembly and surface MHC class I expression.

Human MHC class I (human leukocyte antigen class I [HLA class I]) heavy chains are encoded by three highly polymorphic genes: *HLA-A*, *HLA-B*, and *HLA-C*, with *HLA-B* being the most polymorphic of the three. The high polymorphisms enable the presentation of diverse antigens to CD8$^+$ T cells (*Falk et al., 1991*; *Sarkizova et al., 2020*). The polymorphisms also result in divergent assembly, stability, and expression variations among allotypes (*Rizvi et al., 2014*; *Geng et al., 2018*; *Yarzabek et al., 2018*) as well as peptide repertoire differences (*Falk et al., 1991*; *Sarkizova et al., 2020*). Several HLA class I allotypes can assemble independently of tapasin (*Peh et al., 1998*; *Williams et al., 2002*; *Rizvi et al., 2014*), a key PLC component (*Blum et al., 2013*). Tapasin is known to edit the HLA class I peptide repertoire toward high-affinity sequences (*Rizvi and Raghavan, 2006*; *Wearsch and Cresswell, 2007*; *Chen and Bouvier, 2007*), and tapasin-deficient cells have lower cell surface HLA class I stability than their wild type counterparts (*Garbi et al., 2003*). Allotypes that can assemble independently of tapasin may generally contain suboptimal peptide repertoires, resulting in complexes that are less stable (more rapidly endocytosed; *Zagorac et al., 2012*) and more peptide receptive on the cell surface (a phenotype induced by suboptimal peptide loading; *Ljunggren et al., 1990*; *Schumacher et al., 1990*). Additionally, some HLA-B allotypes are mismatched with TAP in their peptide-binding specificities, which can contribute to suboptimal peptide loading in the ER and resulting reduction in cell-surface stability and expression levels (*Yarzabek et al., 2018*).

Following the expression on the cell surface via the secretory pathway, MHC class I molecules are endocytosed, sorted, and recycled to the cell surface or trafficked to lysosomes for degradation. These are well-studied processes that are likely to be important in antigen presenting cells (APC), such as dendritic cells that have specialized endosomal pathways (*Montealegre and van Endert, 2018*). While some studies have provided evidence for endosomal recycling and assembly in constitutive HLA class I induction (*MacAry et al., 2001*), the major immunologically relevant role for HLA class I recycling is thought to be in antigen cross-presentation, which involves the presentation of exogenous antigens via MHC class I molecules. Endosomal/phagosomal digestion of exogenous antigens and assembly with recycling class I molecules (the vacuolar pathway) is one pathway for HLA class I assembly during cross-presentation (*Colbert et al., 2020*). While MHC class I endosomal recycling has been well studied in APC in the context of cross-presentation, there is little data on the contributions of endosomal assembly to constitutive HLA class I surface expression, and on whether HLA class I polymorphisms result in different efficiencies of or dependencies on endosomal assembly. The conditions for peptide binding within endosomal and phagosomal compartments are quite different compared to canonical assembly in the ER due to lower compartmental pH within endo-lysosomes and the lack of ER chaperones and PLC components. Based on the marked variations in ER assembly characteristics of HLA class I allotypes (*Peh et al., 1998*; *Williams et al., 2002*; *Rizvi et al., 2014*; *Geng et al., 2018*) and known variabilities in cell surface stability and peptide occupancy among HLA-B allotypes (*Yarzabek et al., 2018*), we hypothesized differences in the outcomes of endocytic trafficking and assembly in HLA class I allotype and cell type-dependent manners. This was further investigated using primary human monocytes and monocyte-derived dendritic cells (moDCs).

## Results
### Variable effects of endo-lysosomal pH disruption on surface expression of HLA-B allotypes in moDCs
We examined whether the disruption of endo-lysosomal pH with bafilomycin, an inhibitor of the V-ATPase responsible for maintaining the endo-lysosomal pH gradients (*Forgac, 2007*; *Lafourcade et al., 2008*), would differentially affect the cell surface expression levels of individual HLA-B allotypes. All

HLA-B allotypes can be grouped into either the HLA-Bw4 or HLA-Bw6 serotypes, which are recognized by specific antibodies. For the first set of analyses, donors who express one HLA-B-Bw6 and one HLA-B-Bw4 were chosen. Whereas some HLA-A allotypes are recognized by anti-Bw4, donors were selected such that no cross-reactive HLA-A allotypes were expressed (*Yarzabek et al., 2018*). Some HLA-C allotypes are also recognized by anti-Bw6 (*Yarzabek et al., 2018*), but donor selection ensured the absence of more than one cross-reactive HLA-C for each HLA-B with a Bw6 epitope (Groups 1–8, *Supplementary file 1*). Previous studies suggest that HLA-C protein expression in peripheral blood mononuclear cells (PBMCs) is at least four- to sixfold lower than HLA-B (*Apps et al., 2015*). The significant dominance of HLA-B protein expression relative to HLA-C protein expression (>fourfold for surface and >threefold for total) was confirmed by comparisons of anti-HLA-Bw6 signals in moDCs from three sets of donors expressing either HLA-B or HLA-C allotypes with Bw6 epitopes, but not both (*Figure 1—figure supplement 1*). Thus, in the selected donors (Groups 1–8, *Supplementary file 1*), the dominant signals measured by anti-Bw6 or anti-Bw4 arise from individual HLA-B allotypes of interest, as previously described (*Yarzabek et al., 2018*).

Endosomal pH disruption by bafilomycin has been shown previously to slow receptor recycling to the surface, but not internalization (*Johnson et al., 1993*). The responses of four common HLA-Bw6 allotypes (B*07:02, B*08:01, B*15:01, and B*35:01, *Figure 1A*) and four common HLA-Bw4 allotypes (B*27:05, B*51:01, B*44:02, and B*57:01, *Figure 1B*) to a 4-hr time course of bafilomycin treatment were examined in moDCs. The HLA-B allotypes display a range of sensitivities to bafilomycin for their cell surface expression. Some allotypes, such as B*08:01 and B*44:02, largely resist downregulation by bafilomycin, while others, such as B*35:01 and B*57:01, are highly sensitive to downregulation (*Figure 1C*). B*35:01 is consistently highly sensitive to bafilomycin treatment, aligning with its better capability for tapasin and TAP-independent assembly (*Figure 1D and E*). However, relative PLC independence does not fully explain the bafilomycin trends, as the highly PLC-dependent allotype B*57:01 is similarly sensitive to bafilomycin treatment as B*35:01. B*57:01 is distinguished by its relatively diverse peptidome (*Figure 1F* and *Yarzabek et al., 2018*), and furthermore, abacavir, an anti-HIV drug, is able to permeate into the peptide-binding site of B*57:01 and alter its peptide repertoire (*Illing et al., 2012*). These and other characteristics of B*57:01 could underlie a superior peptide exchange potential of B*57:01. Finally, internalization of HLA class I molecules is dependent on the cytoplasmic tail (*Capps et al., 1989*), and for HLA-A and HLA-B molecules, is specifically modulated via a cryptic sorting motif with the conserved tyrosine $Y^{320}$ (*Le Gall et al., 1998*). However, there are no significant differences between the tails of B*08:01, B*35:01, B*44:02, and B*57:01, other than a $C^{325}S$ polymorphism present in B*35:01 (*Figure 1G*). It is unknown whether this variation at AA 325 affects HLA class I endocytosis and sorting. While multiple mechanisms could contribute to the variable bafilomycin sensitivity of HLA-B allotypes, overall, the analyses of *Figure 1A–C* reveal that a subset of HLA-B allotypes are more reliant on proper endo-lysosomal pH and trafficking for maintaining their constitutive cell surface expression in moDCs.

To better understand the overall pattern of HLA class I sensitivity/resistance to bafilomycin in moDCs, we also examined changes to the cell surface expression of $\beta_2$m, the soluble light chain of MHC class I heterodimers. The BBM.1 antibody recognizes an epitope on $\beta_2$m (*Parham et al., 1983*), and, as $\beta_2$m is a soluble protein, it requires associations with classical or non-classical MHC class I heavy chains to be detectable on the cell surface. Despite considerable donor-dependent variations, bafilomycin treatment had an overall significant negative effect on cell surface BBM.1 levels in moDCs (*Figure 1H*), suggesting that the assembly of multiple classical/non-classical MHC class I allotypes is negatively impacted by bafilomycin treatment. On the other hand, this was not accompanied by a parallel significant induction of the cell surface signal for the HC10 antibody (*Figure 1I*), which detects peptide-deficient heavy chain conformations (*Stam et al., 1990*), indicating that, in general, the loss of cell surface $\beta_2$m is coincident with internalization of heterodimeric HLA class I complexes.

## Low cell-surface HLA-Bw6 half-lives in moDCs and allotype-dependent differences in peptide receptivity

To understand the mechanisms and consequences of varying bafilomycin sensitivity of cell-surface HLA-B in moDCs, further investigations were focused on B*08:01 vs B*35:01, both Bw6 allotypes, but at opposite ends of bafilomycin sensitivity (*Figure 1*). We first determined the cell surface half-lives of HLA-Bw6 on the surface of B*35:01⁺ or B*08:01⁺ moDCs by treating with brefeldin A (BFA) and

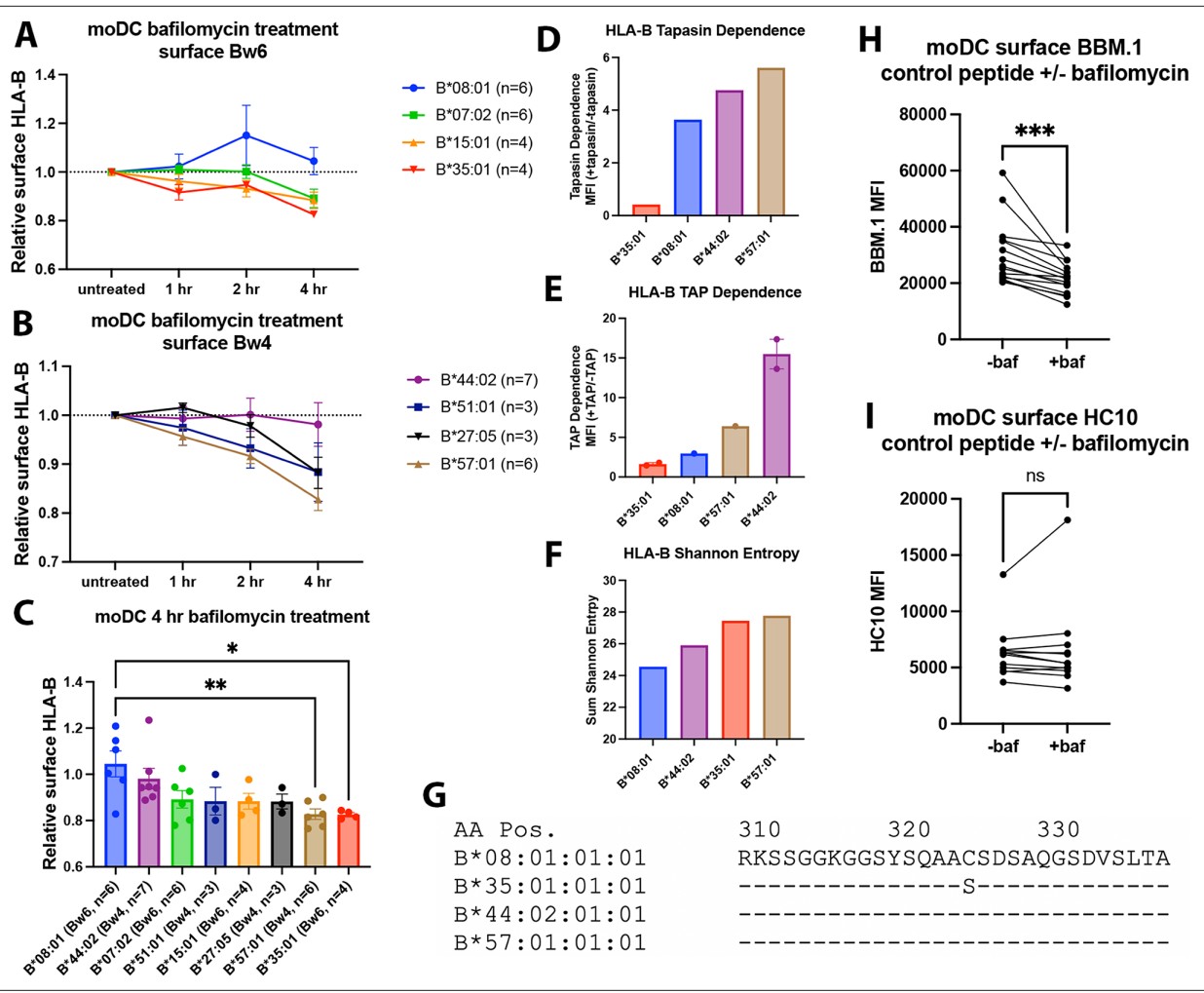

**Figure 1.** Varying human leukocyte antigen B (HLA-B) dependencies on endo-lysosomal pH for surface expression on monocyte-derived dendritic cells (moDCs). (**A and B**) moDCs from select donors (*Supplementary file 1*) were treated with bafilomycin A1 over a 4-hr time course and stained with monoclonal anti-Bw6 (**A**) or anti-Bw4 (**B**) antibodies to measure surface HLA-B. (**C**) HLA-B expression, assessed with either anti-Bw6 or anti-Bw4, after 4 hr of bafilomycin treatment was compared across allotypes with a one-way ANOVA. B*08:01+ donors (n=6 experiments): 55 (n=2), 94 (n=1), 166 (n=1), 178 (n=1), and 198 (n=1); B*44:02+ donors (n=7 experiments): 94 (n=1), 128 (n=2), 196 (n=1), and 267 (n=3); B*07:02+ donors (n=6 experiments): 14, 20, 269 (n=3), and 270 (n=1); B*51:01+ donors (n=3 experiments): 14 (n=1) and 232 (n=2); B*15:01+ donors (n=4 experiments): 124 (n=2) and 128 (n=2); B*27:05+ donors (n=3 experiments): 142, 250, and 256; B*57:01+ donors (n=6 experiments): 156 (n=1), 178 (n=1), 210 (n=1), and 269 (n=3); B*35:01+ donors (n=4 experiments): 24 (n=2), 187 (n=1), and 210 (n=1). (**D**) Relative tapasin dependence of B*08:01, B*35:01, B*44:02, and B*57:01, quantified by the surface expression in M553 cells with or without tapasin. Data are from *Rizvi et al., 2014*. (**E**) Relative transporter associated with antigen processing (TAP) dependence of four HLA-B allotypes, quantified by the ratio of surface expression in the presence or absence of TAP. The HLA-B expression was measured with the W6/32 antibody (B*08:01, B*35:01, and B*44:02), anti-HA antibody (B*35:01, B*57:01, and B*44:02) or both (B*44:02 and B*35:01). Data are from *Geng et al., 2018*. (**F**) HLA-B peptidome diversity is assessed by calculating the Shannon Entropy values at the $P_1$-$P_5$ and $P_C$-$P_{C-2}$ amino acid positions of 8–11 mer peptides identified for each allotype from the immunopeptidome analyses of *Sarkizova et al., 2020*. The Shannon Entropy calculations were performed as described by *Yarzabek et al., 2018*. For each allotype, the sum of the Shannon Entropy values for the $P_1$-$P_5$ and $P_C$-$P_{C-2}$ positions is calculated and plotted for 8–9 mer peptides (B*08:01) or 9–11 mer peptides (other allotypes). (**G**) The cytoplasmic tail regions of four HLA-B allotypes, starting at amino acid 310, were aligned using sequences from the Immuno Polymorphism Database (https://www.ebi.ac.uk/ipd/imgt/hla/). (**H**) moDCs were pulsed with either B*08:01 or B*35:01 control peptides (mutated canonical peptides with poor binding) in the presence or absence of bafilomycin for 4 hr. Cells were stained with the anti-β2m antibody BBM.1. N=15 experiments, significance ± bafilomycin was assessed using paired t tests. (**I**) Experiments were performed as in (**H**) but stained with the antibody HC10. N=12 experiments, significance ± bafilomycin was assessed using paired t tests.

The online version of this article includes the following source data and figure supplement(s) for figure 1:

**Source data 1.** Monocyte-derived dendritic cell (moDC) human leukocyte antigen B (HLA-B) bafilomycin time course.

**Source data 2.** Human leukocyte antigen B (HLA-B) tapasin-dependence, transporter associated with antigen processing-dependence, and Shannon

*Figure 1 continued on next page*

*Figure 1 continued*

Entropy.

**Source data 3.** BBM.1 and HC10 staining with and without bafilomycin.

**Figure supplement 1.** Human monocyte-derived dendritic cells (moDCs) express human leukocyte antigen B (HLA-B) at least four times higher than HLA-C on the cell surface, as detected with anti-Bw6.

**Figure supplement 1—source data 1.** Relative anti-Bw6 cross-reactivity to human leukocyte antigen C (HLA-C) on monocyte-derived dendritic cells.

measuring the surface decay-rates over a 4-hr time course. BFA treatment blocks forward trafficking of proteins from the ER to the cell surface by disrupting the Golgi complex *Fujiwara et al., 1988*; correspondingly, this assay measures how rapidly HLA-B is endocytosed from the cell surface (*Figure 2A*). B*08:01$^+$ moDCs have a trend towards a longer Bw6 half-life compared to B*35:01$^+$ moDCs, in line with our previous measurements in lymphocyte subsets (*Yarzabek et al., 2018*; *Figure 2B*). Tapasin-independent assembly has been linked to reduced thermostability for the tapasin-independent HLA-B*44:05 compared with the highly tapasin-dependent B*44:02 allotypes in tapasin-sufficient B-lymphobastoid cell line (*Williams et al., 2002*), when assessed with the conformation-sensitive pan-HLA class I antibody W6/32 (*Barnstable et al., 1978*). When measured with anti-Bw6, although there is a trend toward decreased thermostability of B*35:01 compared with B*08:01 following 2 hr of 42°C exposure, this did not reach significance (*Figure 2C*). Also notable is the increase in surface expression of B*08:01 but not B*35:01 following 1 hr at room temperature (RT; *Figure 2C*). While more studies are needed to understand this increase, it is possible that at low temperatures, more B*08:01 is able to escape ER quality control to become expressed on the cell surface (*Song et al., 1999*).

We next employed a peptide receptivity assay previously developed by our lab (*Yarzabek et al., 2018*) to compare peptide loading differences between HLA-B allotypes. The HC10 antibody (*Stam et al., 1990*) that recognizes 'open' HLA class I was used. To verify that this antibody can be used to measure peptide receptivity and loading for both allotypes, we performed a bead-based in vitro assay. Purified B*08:01 and B*35:01 monomers pre-loaded with low-affinity peptides (DAN or APL, respectively) were incubated with allotype-specific peptides of medium- (GPK and EPL, respectively, for B*08:01 and B*35:01) or high affinity (FLR and HPV, respectively, for B*08:01 and B*35:01). Both allotypes efficiently exchange the high-affinity peptide at pH 6–7, indicated by the reduction in HC10 signal (*Figure 2D*). Significantly smaller but measurable reductions in HC10 signals were also observed with the low-affinity peptides. Thus, HC10 is responsive to peptide loading by both B*08:01 and B*35:01.

To assess B*08:01 and B*35:01 peptide receptivity in moDCs, their specific peptides were incubated with relevant donor cells for 4 hr. Additionally, peptides that were truncated at the C terminus and mutated at key anchor residues were used as non-binding-specific control peptides. Peptide binding to the moDC HLA-B was measured as the reduction in HC10 signal by specific peptide relative to control peptide (*Yarzabek et al., 2018*). When moDCs were pulsed with two sets of specific and control peptides, B*35:01 was in general receptive to peptide, whereas B*08:01 was not (*Figure 2E*). A similar trend was previously observed in lymphocytes (*Yarzabek et al., 2018*) that have greater HLA class I half-lives compared with moDCs (see Discussions below); thus, more rapid internalization of HLA-Bw6 in moDCs compared to other leukocytes is not explained by suboptimal peptide loading per se. Additionally, B*35:01 peptide receptivity on moDCs is somewhat temperature sensitive, as incubation with peptide at 4°C reduced the binding (*Figure 2F*). This finding indicates that at least a portion of the peptide binding may occur in the endosomal compartments, since 4°C incubation inhibits endocytic trafficking. Additionally, peptide binding may occur in the trans-Golgi network (TGN) or Golgi apparatus, as peptide-receptive MHC class I conformers have been observed in the Golgi (*Day et al., 1995*), corresponding with observed endosome-to-TGN retrograde recycling pathways (*Johannes and Popoff, 2008*). However, B*35:01 is also more peptide receptive at 4°C than B*08:01, indicating that both on the surface and within intracellular compartments, B*35:01 can bind exogenous peptide more efficiently than B*08:01.

Overall, the findings of *Figures 1 and 2* suggest that increased peptide receptivity of moDC HLA-B*35:01 is linked to a greater propensity for assembly with endogenous endosomal peptides and that blockage of endosomal acidification with bafilomycin interferes with such assembly, resulting in reduced cell surface expression.

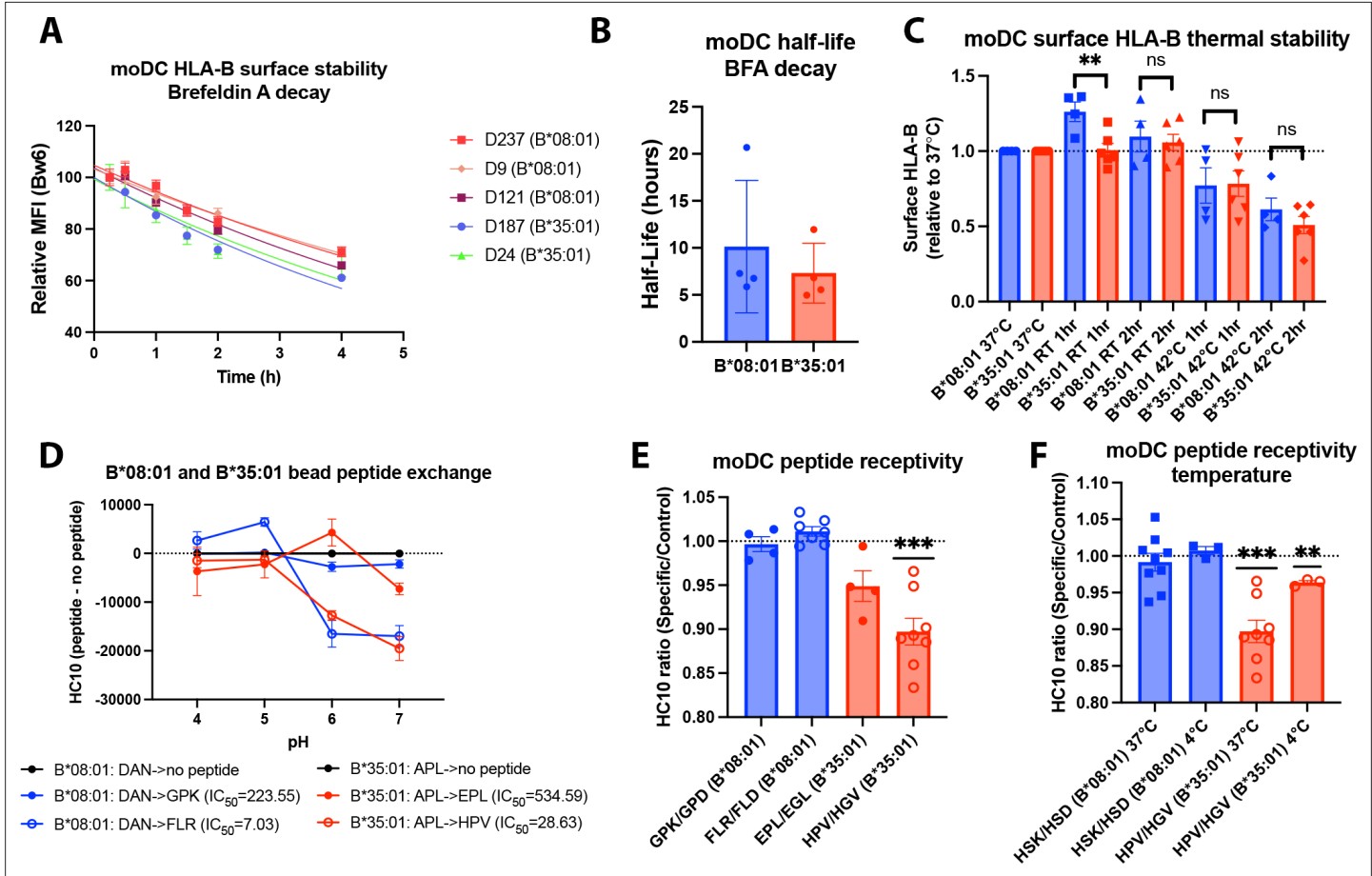

**Figure 2.** Assessments of allotype-dependent differences in (human leukocyte antigen-Bw6) HLA-Bw6 surface stability and peptide occupancy in monocyte-derived dendritic cells (moDCs). (**A**) Representative moDC HLA-Bw6 surface decay plots over a 4-hr time course after treatment with brefeldin A (BFA). (**B**) Average HLA-Bw6 half-life on moDCs extrapolated from BFA decay-rates. B*35:01+ donors were: 24 (n=1) 187 (n=2), and 210 (n=1). B*08:01+ donors were: 9 (n=1), 121 (n=1), 198 (n=1), and 237 (n=1). B*08:01+ n=4 experiments (four donors), B*35:01+ n=4 experiments (three donors). Data were analyzed with an unpaired t test. (**C**) HLA-B thermostability assays were performed by incubating B*08:01+ or B*35:01+ moDCs at room temperature (RT), 37°C, or 42°C for 1 or 2 hr, followed by staining with anti-Bw6 antibody. HLA-B expression was normalized to 37°C, and unpaired t tests were used to assess differences between B*08:01 and B*35:01 responses. B*08:01 donors (n=4 experiments): 28 (n=1), 121 (n=1), 178 (n=1), and 198 (n=1). B*35:01 donors (n=6 experiments): 24 (n=2), 187 (n=1), 210 (n=2), and 274 (n=1). (**D**) Epitope-linked purified and biotinylated B*08:01 and B*35:01 proteins procured from the NIH tetramer core were digested overnight with PreScission protease to release the covalently linked peptides and replace with low-affinity peptides (DAN for B*08:01 and APL for B*35:01). This was followed by binding to streptavidin beads and washing. Exchange with medium- or high-affinity peptides at various pH values was undertaken by incubation for 1.5 hr at 37°C, followed by staining with the HC10 monoclonal antibody. N=2 experiments were undertaken in replicate for each allotype. (**E**) moDCs pulsed with B*08:01-specific peptides GPK and FLR or B*35:01-specific peptides, EPL and HPV, were stained with the monoclonal antibody HC10 to measure peptide-receptive HLA-B. Staining with specific peptides was plotted as a ratio to the control peptides GPD and FLD (for B*08:01) or EGL and HGV (for B*35:01). Ratios were compared to 1 (no difference between specific and control peptides) using a one sample t test. B*08:01 donors were: 9, 94, 105, 121, 130, 148, 166, 178, 198, and 237. B*35:01 donors were: 24, 168, 187, and 210. B*08:01 GPK/GPD n=4, FLR/FLD n=7. B*35:01 HPV/HGV n=8, EPL/EGL n=4. (**F**) moDC peptide receptivity experiments as in (**E**) with HSK/HSD peptide pair for B*08:01 and HPV/HGV peptide pair for B*35:01, including incubation with peptide at 4°C. n=3 experiments were undertaken at 4°C for each allotype.

The online version of this article includes the following source data for figure 2:

**Source data 1.** Monocyte-derived dendritic cell (moDC) human leukocyte antigen B half-life with brefeldin A treatment.

**Source data 2.** Surface human leukocyte antigen B (HLA-B) thermal stability.

**Source data 3.** Human leukocyte antigen B (HLA-B) bead pH peptide exchange (HC10).

**Source data 4.** Human leukocyte antigen B peptide receptivity on monocyte-derived dendritic cells (moDCs).

## Distinct HLA-B distributions and dynamics within endosomal compartments of monocytes and moDCs

To examine and compare endo-lysosomal distributions of HLA-B*35:01 and B*08:01, we conducted confocal microscopy studies, both in moDCs and in their precursor monocytes, and quantified co-localization with three markers: EEA1, a marker of early endosomes, Rab11, a marker of recycling endosomes and a previously described endosomal storage compartment in DCs (*Montealegre and van Endert, 2018*), and LAMP1, a lysosomal marker. HLA class I is known to traffic constitutively through the endosomal system, first entering the EEA1$^+$ early endosomes and from there either recycling through the Rab11$^+$ recycling endosome back to the surface or routing to the lysosome for degradation. Additional trafficking pathways have been described for DCs and other professional antigen-presenting cells, particularly branching from a Rab11$^+$ perinuclear storage compartment (reviewed in *Montealegre and van Endert, 2018*). This compartment has been implicated as a source of HLA class I for vacuolar cross-presentation pathways (*Nair-Gupta et al., 2014*), and thus we hypothesized that Rab11$^+$ endosomes would be a major compartment of HLA-B accumulation and endosomal assembly, particularly for B*35:01.

Monocytes and moDCs were plated onto coverslips, fixed, and stained for HLA-Bw6 along with either EEA1, Rab11, or LAMP1. Representative images for monocytes and moDCs are shown in *Figure 3A and B*, respectively. Monocytes are smaller and rounded compared to moDCs, which have more cytoplasm and some spindle-shaped extensions on the edges. Endocytic HLA-Bw6 in monocytes dominantly localizes to the lysosomes, as both B*08:01$^+$ and B*35:01$^+$ cells have the greatest co-localization with LAMP1 (*Figure 3C and D*). Based on object-based co-localization analyses, monocyte HLA-Bw6 is least co-localized with EEA1. In moDCs, the endocytic HLA-Bw6 is more broadly distributed across each compartment, showing the greatest co-localization with Rab11$^+$ endosomes (*Figure 3E and F*). Similar co-localization trends were observed when the microscopy data was analyzed by the Pearson's correlation method. Monocytes have high HLA-Bw6 co-localization with both Rab11 and LAMP1, whereas moDCs have high HLA-Bw6 co-localization with Rab11 (*Figure 3—figure supplement 1*).

Together, the results of *Figures 2 and 3* indicate that cell-specific differences dictate HLA-Bw6 half-lives and intracellular distributions in monocytes and moDCs. As shown in *Figure 2B*, the average half-life for moDC HLA-Bw6 is about 5 hr. We have previously measured the HLA-Bw6 half-life on monocytes to be in a similar range (*Yarzabek et al., 2018*). In the immature state, moDCs appear predisposed to more rapid internalization of HLA class I compared to some lymphocyte subsets (*Figure 3—figure supplement 2*). On the other hand, monocytes compared to moDC have greater HLA-Bw6 trafficking to lysosomes, indicative of inefficient endosomal assembly and/or more rapid endosomal maturation to lysosomal compartments. Meanwhile, moDCs retain much of their endosomal HLA-Bw6 in Rab11$^+$ endosomes, an essential storage compartment for HLA class I recruitment to DC antigen$^+$ endosomes during cross-presentation (*Montealegre and van Endert, 2018*). More rapid endosomal maturation in monocytes (*Figure 3C–D*) could deplete the endosomal HLA-Bw6 pools, whereas sorting to recycling endosomes (*Figure 3E–F*) could maintain greater HLA-B endosomal sampling in moDCs. Overall, human moDCs are more poised for HLA-Bw6 endosomal assembly and re-expression on the surface compared to monocytes, which fit the more differentiated and specialized state of moDCs.

## Bafilomycin induces moDC B*35:01 accumulation in lysosomes and reduces its peptide occupancy

To further examine and compare HLA-B assembly within endo-lysosomal compartments of monocytes with moDCs, monocytes were treated over a time course with bafilomycin, and surface HLA-B expression was measured by flow cytometry and compared with the results from moDC (*Figure 4A and B*; a subset of the data from *Figure 1A* for B*0801 and B*35:01 are replotted as histograms in *Figure 4B*). Treatment with bafilomycin over 4 hr resulted in a net increase in the surface expression of HLA-Bw6 in both B*08:01$^+$ and B*35:01$^+$ monocytes (*Figure 4C*). These results are in line with the high lysosomal accumulation of HLA-Bw6 observed with confocal microscopy *Figure 3*; thus, it seems that the increase in lysosomal pH with bafilomycin treatment rescues the HLA-B in this compartment from degradation, particularly in B*08:01$^+$ monocytes (*Figure 4C*). Conversely, as noted in *Figure 1*, HLA-Bw6 expression was unchanged in B*08:01$^+$ moDCs after bafilomycin treatment, whereas B*35:01$^+$ moDCs decreased their HLA-Bw6 expression (*Figure 4D*; a subset of the data from *Figure 1A* for

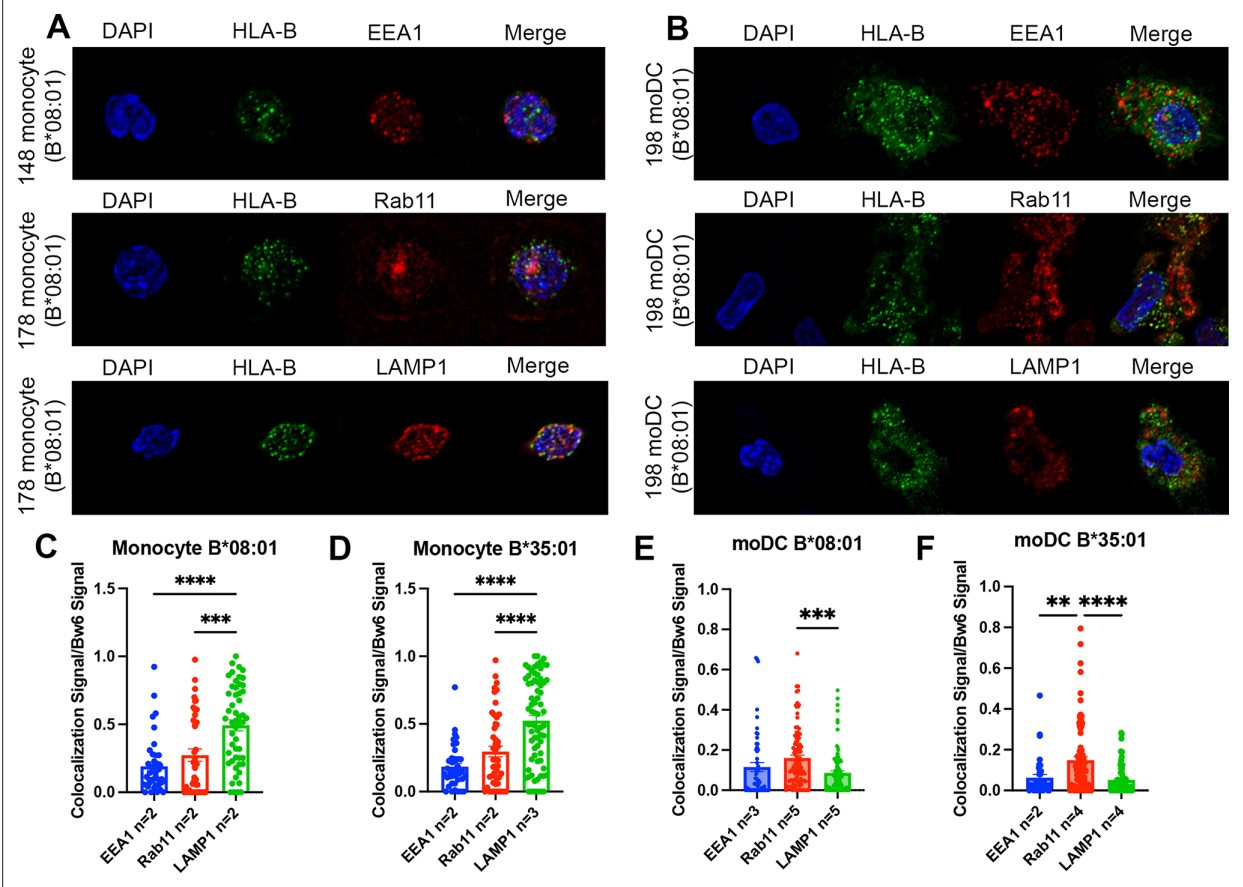

**Figure 3.** Significant accumulation of human leukocyte antigen-Bw6 (HLA-Bw6) in lysosomal (monocytes) or recycling endosomal (monocyte-derived dendritic cells [moDCs]) compartments. (**A**) Representative confocal microscopy images of primary human monocytes stained for HLA-Bw6 co-localization with the early endosome marker EEA1, the recycling endosome marker Rab11, and the lysosomal marker LAMP1. (**B**) Representative moDC staining for HLA-Bw6 co-localization with the same markers. (**C and D**) Monocyte Bw6 co-localization with each indicated marker was quantified by object-based co-localization for two to four B*08:01⁺ or B*35:01⁺ donors. (**E and F**) moDC Bw6 co-localization with each of the indicated marker was quantified by object-based co-localization for B*08:01⁺ or B*35:01⁺ donors. Each point is a cell, with at least 20 individual cells imaged per donor and two to five donors for each co-localization condition. Co-localization data are represented as the fraction of HLA-Bw6 signal overlapping with the second marker signal. Monocyte B*08:01⁺ donors: 55, 130, 148, and 178. Monocyte B *35:01⁺ donors: 24, 136, 187, and 210. moDC B*08:01⁺ donors: 55, 94, 166, 198, and 237. moDC B*35:01⁺ donors: 24, 168, and 187. One-way ANOVAs were used for analysis to compare the co-localization of HLA-B with each marker.

The online version of this article includes the following source data and figure supplement(s) for figure 3:

**Source data 1.** Monocyte and monocyte-derived dendritic cell (moDC) object-based co-localization.

**Source data 2.** Monocyte and monocyte-derived dendritic cell Pearson's co-localization.

**Figure supplement 1.** Pearson's correlation analysis of monocyte and monocyte-derived dendritic cell (moDC) endo-lysosomal co-localization.

**Figure supplement 2.** Human monocyte-derived dendritic cells (moDCs) have lower human leukocyte antigen-Bw6 cell surface stability compared to lymphocytes.

**Figure supplement 2—source data 1.** Cell-type comparison of surface human leukocyte antigen B half-life.

B*08:01 and B*35:01 are replotted in *Figure 4D*). These findings indicate that, unlike in monocytes, low endo-lysosomal pH is important for maintaining B*35:01 surface expression in moDCs. A gradient of V-ATPase subunits is present along the endosomal pathway, with the highest number of active complexes present in the lysosomes and lowest amount present in early endosomes (*Lafourcade et al., 2008*). Thus, it is likely that assembly and trafficking through the entire endo-lysosomal pathway are perturbed by bafilomycin treatment, which negatively affects B*35:01 in moDCs. In contrast, there is reduced proteasomal dependence of HLA-Bw6 expression in B*35:01⁺ monocytes and moDCs compared with the corresponding B*08:01⁺ cells (*Figure 4—figure supplement 1A and B*). These

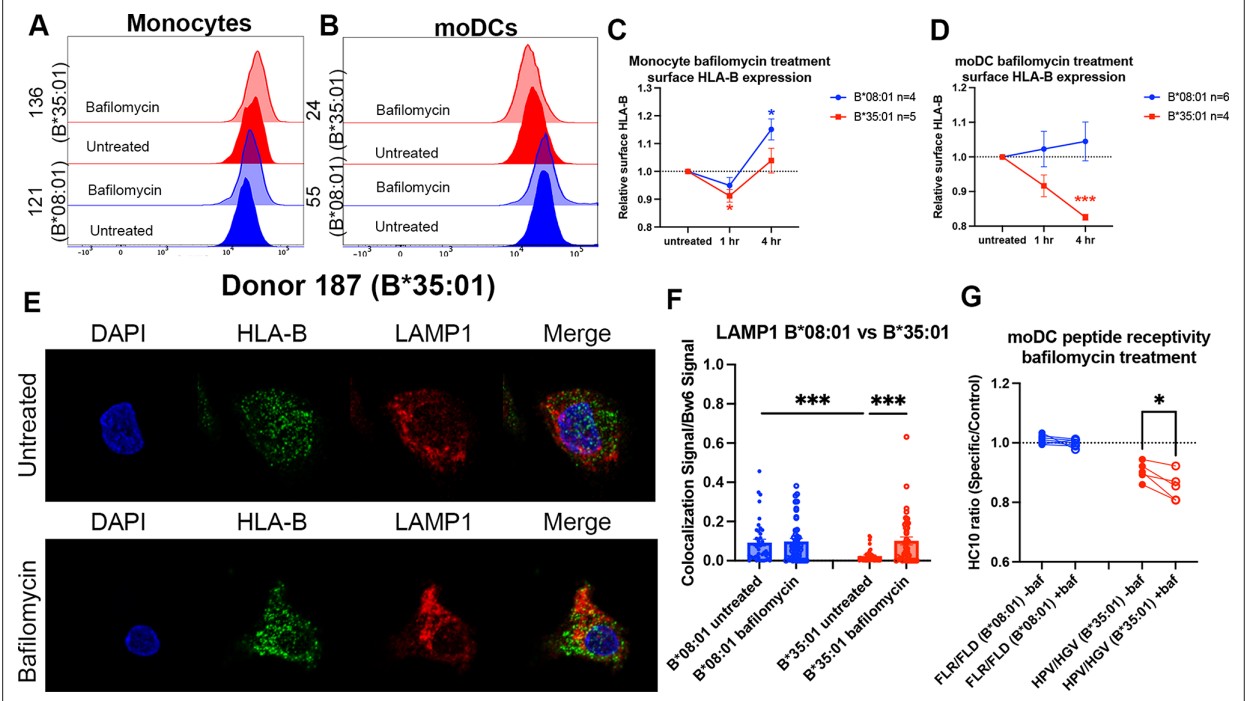

**Figure 4.** Disruptions to endo-lysosomal pH alter human leukocyte antigen B (HLA-B)*35:01 surface expression and induce lysosomal accumulation in monocyte-derived dendritic cells (moDCs). Monocytes or moDCs were treated with 200 nM bafilomycin A1 for 1, 2, or 4 hr. Treatment with bafilomycin was followed by staining for surface markers and HLA-Bw6, followed by analysis via flow cytometry. (**A and B**) Representative HLA-B expression flow cytometry histograms are shown for monocytes (**A**) and moDCs (**B**). (**C**) Relative changes in HLA-Bw6 expression on the surface of HLA-B*08:01+ (n=4) or HLA-B*35:01+ (n=5) monocytes over the 4-hr bafilomycin time course. B*08:01 and B*35:01 expression at each time point was compared to the normalized untreated expression with one sample t tests. B*08:01+ donors for these experiments: 121, 130 (n=2), and 237. B*35:01+ donors for these experiments: 24, 141 (n=2), and 210 (n=2). (**D**) Relative changes in HLA-Bw6 expression on the surface of B*08:01+ (n=6) and B*35:01+ (n=4) moDCs. B*08:01 and B*35:01 expression at each time point was compared to the normalized untreated expression with one sample t tests. B*08:01+ donors for these experiments: 55 (n=2), 94, 166, 178, and 198. B*35:01+ donors for these experiments: 24 (n=2), 187, and 210. (**E**) moDC confocal microscopy experiments comparing HLA-Bw6 co-localization with LAMP1 with and without bafilomycin treatment. (**F**) Object-based co-localization quantification of HLA-Bw6 with LAMP1 with and without bafilomycin treatment. B*08:01+ donors were 94 and 237, and the B*35:01+ donors were 168 and 187 (n=2 for each group). Unpaired t tests were used to compare co-localization with and without bafilomycin. (**G**) Peptide receptivity of B*08:01 and B*35:01 carried out in the presence or absence of bafilomycin. Receptivity of each allotype was compared ±baf treatment via paired t tests. B*08:01+ donors: 9, 105, 121, 130, 148, 166, and 178; n=7 independent experiments. B*35:01+ donors: 24, 168, 187, and 210 (n=2); n=5 independent experiments.

The online version of this article includes the following source data and figure supplement(s) for figure 4:

**Source data 1.** Monocyte and monocyte-derived dendritic cell (moDC) bafilomycin time course.

**Source data 2.** Monocyte-derived dendritic cell (moDC) bafilomycin confocal microscopy.

**Source data 3.** Monocyte-derived dendritic cell (moDC) bafilomycin peptide receptivity.

**Figure supplement 1.** Reduced B*35:01 dependence on proteasomal processing in monocytes and monocyte-derived dendritic cells (moDCs), and additional human leukocyte antigen B (HLA-B) localization/peptide receptivity studies in bafilomycin-treated cells.

**Figure supplement 1—source data 1.** Monocyte and monocyte-derived dendritic cell (moDC) MG132 time course.

**Figure supplement 1—source data 2.** Supplemental confocal co-localization data.

**Figure supplement 1—source data 3.** Human leukocyte antigen B (HLA-B) bead pH peptide exchange (BBM.1).

**Figure supplement 1—source data 4.** Monocyte-derived dendritic cell (moDC) peptide receptivity (BBM.1).

findings suggest that, in line with previously observed tapasin and TAP-independent expression (*Rizvi et al., 2014*; *Geng et al., 2018*; *Bashirova et al., 2020*), B*35:01 compared with B*08:01 is less reliant upon canonical proteasome-dependent ER assembly but more reliant on endosomal assembly.

Next, we treated moDCs with bafilomycin as before, but performed confocal microscopy to examine endo-lysosomal HLA-B redistribution after 4 hr of bafilomycin treatment. Representative images do not show obvious changes in HLA-Bw6 or LAMP1 distribution throughout the cell after treatment (*Figure 4E*). However, quantifications indicate increased HLA-Bw6 co-localization with

LAMP1$^+$ lysosomes upon treatment with bafilomycin in B*35:01$^+$ moDCs, but not in B*08:01$^+$ moDCs (*Figure 4F*). Additionally, in the untreated condition, there is higher steady-state HLA-Bw6 localization to lysosomes in B*08:01$^+$ moDCs compared with B*35:01$^+$ moDCs. Bafilomycin treatment increases HLA-Bw6/LAMP1 co-localization in B*35:01$^+$ moDCs to the B*08:01$^+$ moDC levels. Bafilomycin treatment generally has no effect on HLA-Bw6/Rab11 co-localization in B*35:01$^+$ moDCs (*Figure 4—figure supplement 1C*), and while object-based co-localization analyses indicate increased co-localization of HLA-Bw6 with Rab11 in B*08:01$^+$ moDCs (*Figure 4—figure supplement 1D*), the Pearson's correlation analyses do not support this conclusion (*Figure 4—figure supplement 1E*). Pearson's correlation analyses confirm more significant increases in B*35:01 localization to lysosomes following bafilomycin treatment (*Figure 4—figure supplement 1F*).

Finally, bafilomycin treatment further enhances the peptide receptivity of B*35:01 in moDCs, indicating that bafilomycin reduces the normal level of antigen supply or alters assembly efficiency for B*35:01 (*Figure 4G*). To assess changes to β$_2$m-heavy chain heterodimers in bafilomycin-treated cells, we used the monoclonal antibody BBM.1 as in *Figure 1*. Via a beads-based assay similar to that described in *Figure 2*, peptide binding increased the BBM.1 signal (likely resulting from stabilization of β$_2$m association with heavy chains and beads in the presence of peptides), particularly at more neutral pH values (*Figure 4—figure supplement 1G*). When peptide receptivity experiments were repeated in moDCs with the BBM.1 antibody, in contrast to the HC10-based assays, the addition of specific peptide did not significantly alter the surface BBM.1 signal in B*35:01 donors (*Figure 4—figure supplement 1H*). Additionally, moDC bafilomycin treatment did not change the surface BBM.1 signal in response to specific peptides (*Figure 4—figure supplement 1I*). Based on the considerable global loss of surface BBM.1 signal following bafilomycin treatment (*Figure 1H*), the peptide receptivity of individual allotypes is expected to be harder to detect with BBM.1, which likely recognizes multiple classical and non-classical heavy chain-β$_2$m heterodimers of the moDCs. In contrast, the smaller and specific allotype-induced changes to the HC10 signals (*Figures 1I and 2E*) are more readily detectable.

Together, the data of *Figure 4* indicates that raising the endo-lysosomal pH via bafilomycin treatment results in reduced surface expression and enhanced lysosomal accumulation of B*35:01, which are coincident with accumulation of more peptide-receptive B*35:01 conformers. HLA-Bw6 surface expression is unaffected by bafilomycin in B*08:01$^+$ moDCs. Thus, endosomal pH-dependent processes are required by B*35:01, but not B*08:01, for maintaining optimal constitutive expression in moDCs.

## Monocytes and moDCs differ in exogenous antigen uptake efficiencies and processing pathways

Human moDCs are shown to process and assemble antigens for cross-presentation in their endo-lysosomal (vacuolar) compartments (*Tang-Huau et al., 2018*), a pathway we predicted would be more permissive for B*35:01 based on the results described in *Figures 1–4*. An additional cytosolic pathway of antigen degradation during cross-presentation has also been well characterized and primarily uses the proteasome for peptide processing after antigen translocation from the endosome/phagosome to the cytosol (*Colbert et al., 2020*). Although not much work has been done to investigate antigen processing by monocytes, it is generally believed that differentiation into DCs is required for induction of monocyte cross-presentation function (*Döring et al., 2019*). Consistent with this view, monocytes and moDCs uptake soluble antigen to different extents, with greater moDC endocytosis of fluorescently labeled bovine serum albumin over a 60-min time course (*Figure 5A*). To study processing of endocytosed antigen, we used DQ-Ova as a model for antigen degradation. DQ-Ova is a soluble ovalbumin protein labeled with excess, self-quenching BODIPY fluorophores, which become fluorescent upon proteolytic cleavage of the protein. After DQ-Ova pulsing at either 37 or 4°C for 30 min, the cells were chased in media or media + inhibitors, followed by fixation and flow cytometric analysis. When monocytes were chased with lysosome or proteasome inhibitors and normalized to untreated cells, antigen degradation was completely inhibited by bafilomycin treatment (*Figure 5B*). MG132 inhibition of the proteasome had no effect, while the inhibitor combination inhibited similarly to bafilomycin alone. Thus, monocytes mainly use their lysosomes for exogenous protein degradation.

In moDCs, MG132 inhibition actually increases the degradation of DQ-Ova, likely due to a compensatory enhancement of lysosome-mediated degradation (*Pandey et al., 2007*). In contrast

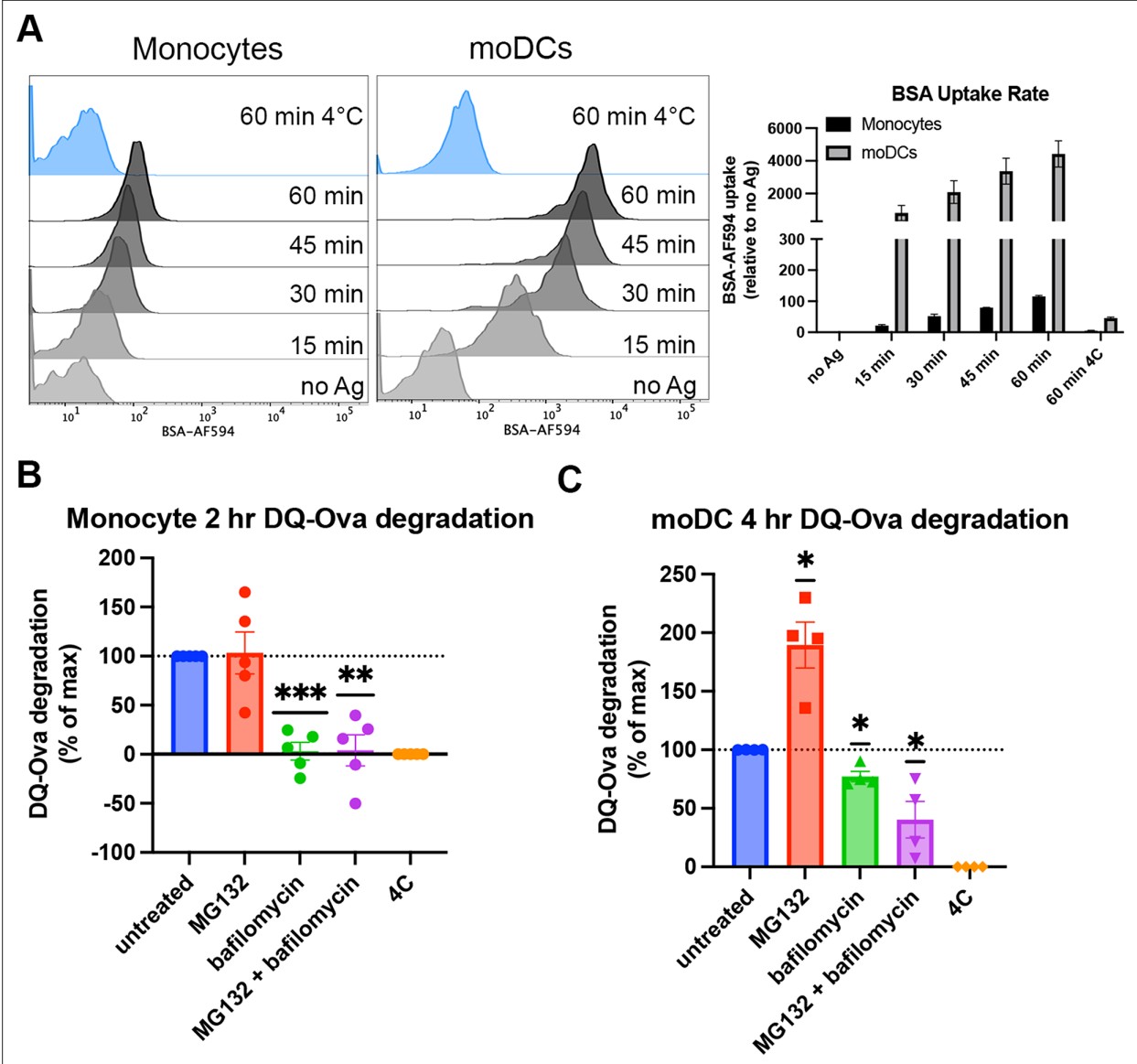

**Figure 5.** Cell type-dependent differences in antigen uptake and processing pathways. (**A**) Monocytes or monocyte-derived dendritic cells (moDCs) were pulsed with BSA labeled with Alexa fluor 594 for 15-min intervals, followed by washing, fixation, and flow cytometric analyses of uptake. Representative histogram plots are shown, as well as averaged uptake rates. N=2 independent experiments for each cell type. Monocyte donors: PCD22F, PCD25F. moDC donors: 255, PCD37M. (**B and C**) Assessments of antigen degradation pathways in monocytes (**B**) and moDCs (**C**) were performed using the soluble antigen DQ-Ova. Monocyte DQ-Ova degradation for 2 hr with inhibitors relative to untreated was quantified in (**B**), and moDC degradation for 4 hr with inhibitors was quantified in (**C**). For monocytes, the experiment was repeated n=5 times, and n=4 times for moDCs. Monocyte donors: 248, 250, 255, 270, and 273. moDC donors: 250, 253, 270, and 275. Data was normalized by subtracting degradation at 4°C from all other conditions and setting degradation at 37°C as the maximum. The effect of each inhibitor on degradation compared to untreated at 37°C was assessed with one sample t test.

The online version of this article includes the following source data for figure 5:

**Source data 1.** Monocyte and monocyte-derived dendritic cell (moDC) antigen uptake.

**Source data 2.** Monocyte and monocyte-derived dendritic cell (moDC) pathways of exogenous antigen processing.

to monocytes, bafilomycin did not completely block DQ-Ova degradation, and the combination of MG132 and bafilomycin further reduced the degradation (*Figure 5C*). As bafilomycin increases the lysosomal pH and inhibits most pH-sensitive proteases present in this compartment, the upregulation of lysosomal degradation by MG132 treatment cannot increase the antigen degradation in the MG132 + bafilomycin combination treatment as seen with MG132 alone. Thus, for soluble protein

degradation, moDCs use both cytosolic and lysosomal pathways. Additionally, the extent of lysosomal degradation of antigen differs between moDCs and monocytes, as bafilomycin alone inhibits monocyte antigen degradation to a greater extent than moDCs. Increased uptake and reduced proteolysis within the endo-lysosomal compartments upon moDC differentiation could explain why some endocytosed antigen undergo proteasomal processing in these cells, as more protein may be preserved for export from endosome to cytosol.

## Cross-presentation via B*35:01 is more efficient than B*08:01 even when matched for T cell response sensitivity and is more affected by cathepsin inhibition

Since endo-lysosomal antigen degradation occurs in both monocytes and moDCs, we further examined the model that B*35:01 could have cross-presentation advantages in both monocytes and moDCs due to its increased propensity both for constitutive endo-lysosomal assembly (*Figures 1 and 4*) and assembly with exogenous peptides (*Figure 2*). To perform cross-presentation assays with human cells, we took advantage of the fact that memory cytotoxic T lymphocytes (CTLs) specific for Epstein-Barr virus (EBV) antigens are broadly prevalent in humans. Two EBV proteins contain known epitopes for both B*08:01 (RAKFKQLL [RAK] and FLRGRAYGL [FLR]) and B*35:01 (EPLPQGQLTAY [EPL] and YPLHEQHGM [YPL]). The FLR and YPL epitopes are derived from the EBNA3A protein, and the RAK and EPL epitopes are derived from the BZLF1 protein (*Thomson et al., 1995*; *Rist et al., 2015*). We sorted and expanded antigen-specific CTLs with B*08:01-RAK, B*08:01-FLR, B*35:01-EPL, and B*35:01-YPL tetramers from donors expressing the relevant HLA-B allotypes (Group 9; *Supplementary file 1*). Peptide titration experiments with the CTLs demonstrated varied sensitivities of each CTL line to peptide, with B*35:01-YPL eliciting the most sensitive response and B*08:01-RAK the least sensitive (*Figure 6A*). B*08:01-FLR and B*35:01-EPL CTLs displayed similar sensitivities to peptide. For a given HLA class I allotype, the epitopes with lower response sensitivities have relatively lower predicted affinities for HLA class I (B*08:01-RAK, $IC_{50}$ 366 nM and B*35:01-EPL, $IC_{50}$ 534 nM) compared with the epitopes with the higher response sensitivities, which have higher predicted affinities (B*08:01-FLR, $IC_{50}$ 7 nM and B*35:01-YPL, $IC_{50}$ 19 nM). This is likely due to lower cell surface levels achieved of the lower-affinity complexes. Consistent with the greater peptide receptivity for B*35:01 (*Figure 2E*), the T cell response sensitivity achieved with the lower-affinity B*35:01-EPL epitope was similar to that achieved with the higher-affinity B*08:01-FLR epitope (*Figure 6A*).

To control for donor-to-donor antigen uptake and processing differences during cross-presentation assays, monocytes and moDCs were used from donors expressing both B*08:01 and B*35:01 so that antigen presentation via both allotypes occurs within the same cells (Group 10; *Supplementary file 1*). The FLR, RAK, YPL, and EPL peptides were used as positive controls for CTL activation, and the purified recombinant EBNA3A or BZLF1 proteins were used as model soluble antigens for cross-presentation (*Figure 6B*). CTL activation was measured by surface CD107a expression (a marker of CTL degranulation) and intracellular IFNγ production. CTL peptide activation assays demonstrated that B*08:01-restricted CTLs were not cross-reactive for B*35:01 peptide and vice-versa (*Figure 6—figure supplement 1*).

As noted above, resting CD14[+] monocytes are generally considered not to be competent for cross-presentation (*Döring et al., 2019*), instead requiring differentiation to moDCs. Surprisingly, however, there is a readily detectable CTL activation in response to EBNA3A cross-presentation, for both the B*08:01-FLR and B*35:01-YPL epitopes (*Figure 6C*). In monocytes, despite the higher percentage of B*08:01-FLR CTL activation in response to peptide (*Figure 6—figure supplement 2A*), B*35:01-YPL CTLs were activated to a greater extent with EBNA3A (*Figure 6—figure supplement 2B*). Normalization of EBNA3A-induced activation to peptide-induced activation confirmed the trend of greater B*35:01-YPL cross-presentation efficiency (*Figure 6D*). These patterns were exaggerated for moDC presentation of peptide and EBNA3A antigen (*Figure 6—figure supplement 2C and D*), where B*35:01-YPL cross-presentation was greatly enhanced compared to monocytes (*Figure 6C*), approaching the level of peptide-induced activation (*Figure 6E*). Thus, in both monocytes and moDCs, the EBNA3A YPL epitope is cross-presented via B*35:01 more efficiently than the FLR epitope via B*08:01.

As the B*35:01-YPL response was sensitive to very low doses of peptide (*Figure 6A*), we sought to confirm our cross-presentation findings with the BZLF1 antigen, which contains RAK, the B*08:01

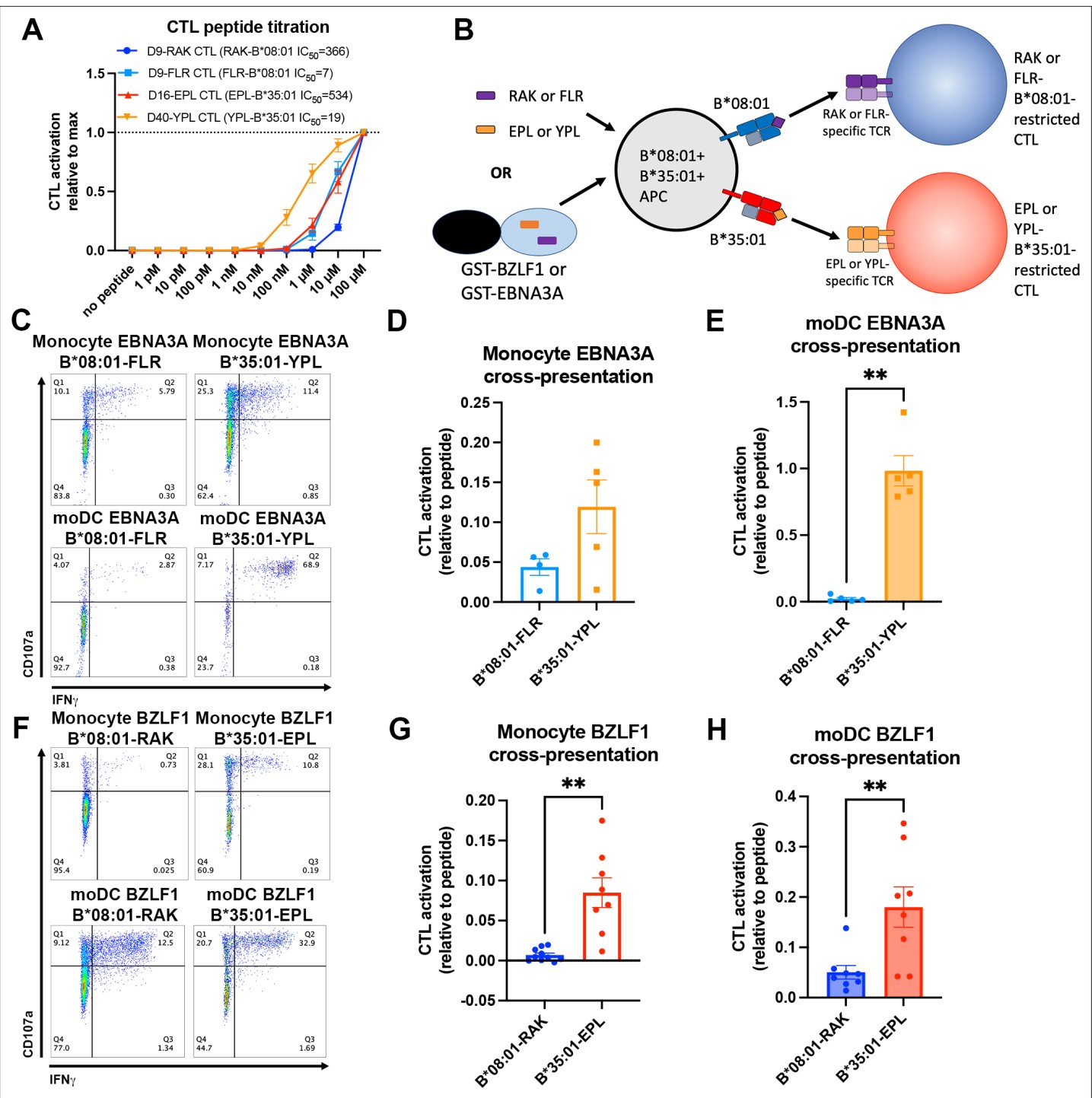

**Figure 6.** Cross-presentation of epitopes derived from Epstein-Barr virus proteins by B*35:01 compared to B*08:01. (**A**) B*08:01-RAK cytotoxic T lymphocytes (CTLs) from donor 9, B*08:01-FLR CTLs from donor 9, B*35:01-EPL CTLs from donor 16, and B*35:01-YPL CTLs from donor 40 were used in peptide titration experiments to measure sensitivity to peptide. Peripheral blood mononuclear cells (PBMCs) from B*08:01+ or B*35:01+ donors were pulsed with peptide overnight at different concentrations, followed by co-culture with each CTL and flow cytometric assessment of activation. B*08:01+ PBMC donors: 94 (n=2) and 148 (n=2). B*35:01+ PBMC donors: 24 (n=4). N=4 independent experiments for B*08:01-RAK and B*35:01-EPL CTLs, and n=3 experiments for B*08:01-FLR and B*35:01-YPL CTLs. (**B**) Schematic representation of cross-presentation assay. B*08:01+/B*35:01+ double-positive monocytes or monocyte-derived dendritic cells (moDCs) were pulsed with either 50 μM canonical B*08:01 peptide (RAK or FLR), 50 μM canonical B*35:01 peptide (EPL or YPL), or purified GST-BZLF1 or GST-EBNA3A protein antigen (100 μg) for 6 hr, then co-cultured with previously expanded CTLs (either B*08:01-restricted or B*35:01-restricted) at a 1:1 CTL:APC ratio for 5 hr. CTLs were assessed for activation by surface CD107a expression and intracellular IFNγ. (**C**) Representative flow cytometry plots are shown for CTLs co-cultured with monocytes and moDCs during EBNA3A

*Figure 6 continued on next page*

*Figure 6 continued*

cross-presentation. CD107a degranulation and intracellular IFNγ expression were measured by flow cytometry. (**D**) Monocyte cross-presentation of EBNA3A quantified as a ratio relative to peptide, n=4 B*08:01-FLR experiments, n=5 B*35:01-YPL experiments. (**E**) moDC cross-presentation of EBNA3A quantified as a ratio relative to peptide, n=5 experiments. (**F**) Representative flow cytometry plots are shown for CTLs co-cultured with monocytes and moDCs during EBNA3A cross-presentation. (**G**) Monocyte cross-presentation of BZLF1 quantified as a ratio relative to peptide, n=9 experiments. (**H**) moDC cross-presentation of BZLF1 quantified as a ratio relative to peptide, n=7 experiments. B*08:01 and B*35:01 cross-presentation in D, E, G, and H compared with paired t tests. Monocyte and moDC APC donors were: 16, 25, and 132. B*08:01-RAK CTL donors: 9 and 16. B*08:01-FLR CTL donor: 9. B*35:01-EPL CTL donor: 16. B*35:01-YPL CTL donor: 40.

The online version of this article includes the following source data and figure supplement(s) for figure 6:

**Source data 1.** Cytotoxic T lymphocyte (CTL) peptide titration.

**Source data 2.** Monocyte and monocyte-derived dendritic cell (moDC) cross-presentation of BZLF1 and EBNA3A.

**Figure supplement 1.** Antigen-specific cytotoxic T lymphocyte (CTL) activation is not cross-reactive within the same antigen source.

**Figure supplement 2.** Peptide and soluble antigen presentation by monocytes and monocyte-derived dendritic cells (moDCs).

**Figure supplement 2—source data 1.** Peptide and soluble Ag percentage of cytotoxic T lymphocyte (CTL) activation.

epitope, and EPL, the B*35:01 epitope. As these are both lower-affinity peptides for the respective HLA-B compared with FLR and YPL, the peptide activation of both BZLF1 epitopes is less sensitive (*Figure 6A*), offering conditions to examine cross-presentation differences when lower cell-surface epitope expression is expected to be achieved. Cross-presentation of the RAK epitope was minimal in monocytes but more detectable in moDCs, while the EPL epitope was more readily cross-presented in both cell types (*Figure 6F*). Although there are no allotype-dependent differences in peptide-mediated activation in monocytes (*Figure 6—figure supplement 2E*), the B*35:01-EPL epitope from the whole BZLF1 antigen activates CTLs to a greater extent than the B*08:01-RAK epitope within the same experiments and same donor APC (*Figure 6—figure supplement 2F*). The B*35:01-EPL advantage for cross-presentation persists when normalized to peptide activation levels within each experiment (*Figure 6G*). In moDCs, there is greater activation of B*35:01-restricted CTLs with both peptide and BZLF1 compared to B*08:01 (*Figure 6—figure supplement 2G and H*), and when BZLF1 activation is normalized to peptide activation, the advantage of B*35:01-EPL CTLs over B*08:01-RAK CTLs for cross-presentation persists (*Figure 6H*).

In comparing the relative cross-presentation efficiencies of all four epitopes in monocytes and moDCs, it is notable that the epitope with the lowest response sensitivity (B*08:01-RAK, *Figure 6A*) displays the lowest cross-presentation efficiency in monocytes, whereas the epitope with the highest response sensitivity (B*35:01-YPL, *Figure 6A*) displays the highest cross-presentation efficiency in moDCs (*Figure 7A and B*). Notably, however, in comparing epitopes with similar response sensitivities (B*08:01-FLR vs B*35:01-EPL; *Figure 7A*), the B*35:01 cross-presentation advantage over B*35:01 is still apparent, particularly in moDCs (*Figure 7C*). The data of *Figures 6 and 7A–C* show that, with intact protein-derived epitopes as with the peptides, B*35:01 peptide loading during cross-presentation is more efficient than for B*08:01.

We further tested the sensitivities of each CTL response to MG132 inhibition of the proteasome, which probes the relevance of the cytosolic pathway of antigen processing and presentation, and to treatment with the cathepsin inhibitor 1 (CTI1), an inhibitor of cathepsins B, K, L, and S (*Figure 7D* [B*08:01] and *Figure 7E* [B*35:01]). Lactacystin, another proteasome inhibitor, was additionally included in some experiments. Cross-presentation via B*08:01 generally displayed significant sensitivity to proteasome inhibition by MG132/lactacystin, whereas non-significant effects were measured for the cathepsin inhibitor (*Figure 7D*). In contrast, B*35:01 responses displayed significant sensitivity both MG132/lactacystin and the CTI1 (*Figure 7E*). Although significance was not achieved for cathepsin inhibition of the B*35:01-YPL response in moDCs, there was a trend toward inhibition. The generally greater sensitivity to CTI1 treatment indicates that B*35:01 uses the vacuolar (endo-lysosomal) pathway of cross-presentation in combination with the cytosolic pathway. The ability to use both the vacuolar and cytosolic pathways of assembly for generating B*35:01-EPL epitopes could explain the higher efficiency of B*35:01-EPL cross-presentation relative to B*08:01-FLR in moDCs.

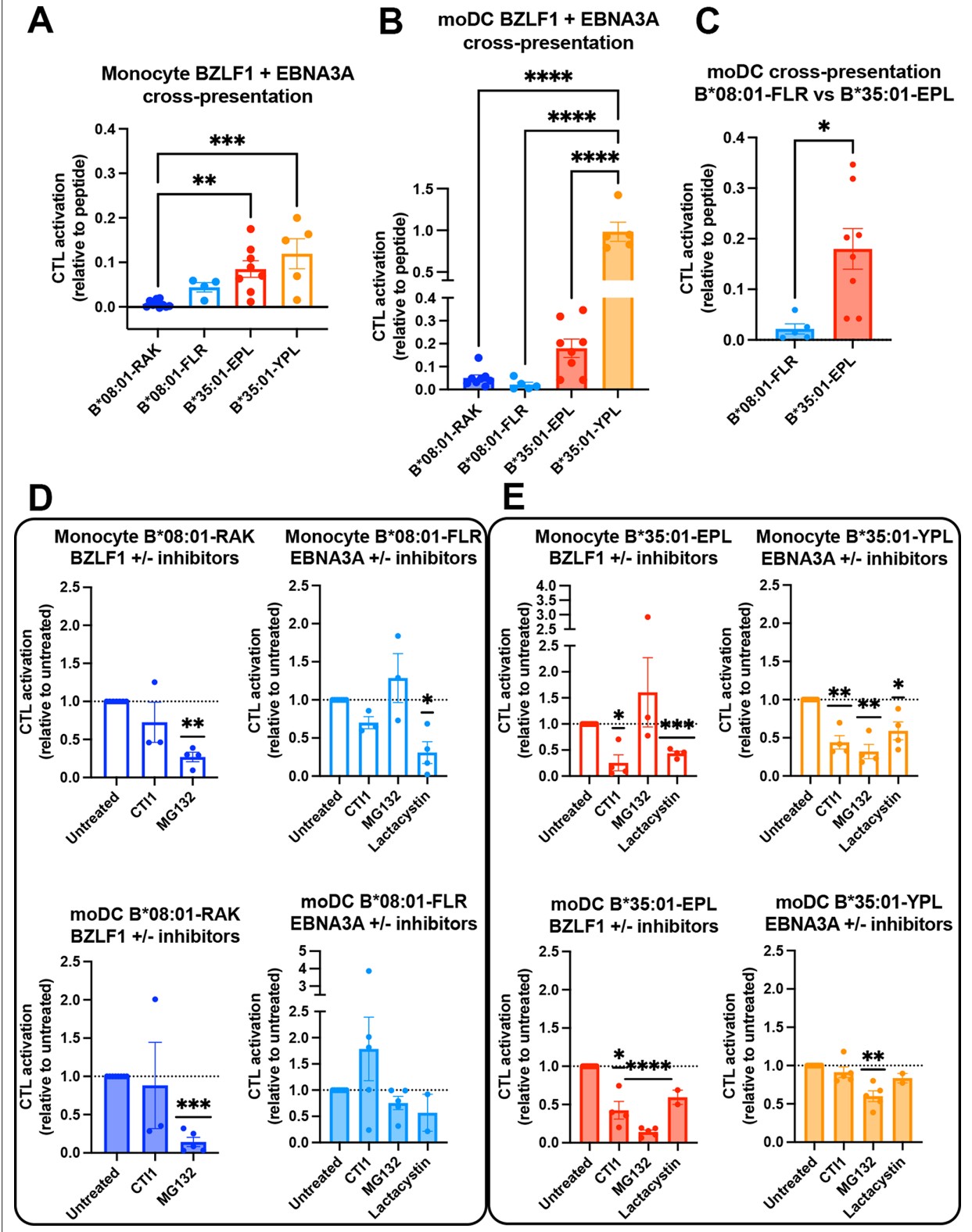

**Figure 7.** Cross-presentation by B*35:01 displays greater sensitivity to cathepsin inhibition and is more efficient than B*08:01 when matched for T cell responsiveness. (**A and B**) Cross-presentation efficiencies of B*08:01-RAK, B*08:01-FLR, B*35:01-EPL, and B*35:01-YPL epitopes from BZLF1 or EBNA3A were compared in monocytes (**A**) or monocyte-derived dendritic cells (moDCs) (**B**). Differences were assessed by one-way ANOVA analysis. (**C**) Two epitopes with similar cytotoxic T lymphocyte (CTL) sensitivities to peptide (*Figure 6A*) were compared for cross-presentation efficiencies in moDCs

*Figure 7 continued on next page*

*Figure 7 continued*

with an unpaired t test, displaying a trend toward more efficient B*35:01-EPL cross-presentation. (**D–E**) Cross-presentation assays were performed as previously described, with the addition of inhibitor treatment. During monocyte or moDC pulse with 100 μg protein antigen, either MG132, lactacystin, or cathepsin inhibitor 1 (CTI1) was added to the antigen presenting cells to inhibit different pathways of antigen processing. Cross-presentation with inhibitors was compared to untreated with a one-sample t test. For monocyte cross-presentation: n=4 MG132 and n=3 CTI1 B*08:01-RAK treatments, n=3 MG132, n=3 CTI1, and n=4 lactacystin B*08:01-FLR treatments, n=3 MG132, n=4 CTI1, and n=4 lactacystin B*35:01-EPL treatments, and n=4 MG132, n=4 CTI1, and n=4 lactacystin B*35:01-YPL treatments. For moDC cross-presentation: n=5 MG132 and n=3 CTI1 B*08:01-RAK treatments, n=5 MG132, n=5 CTI1, and n=2 lactacystin B*08:01-FLR treatments, n=5 MG132, n=4 CTI1, and n=2 lactacystin B*35:01-EPL treatments, and n=5 MG132, n=5 CTI1, and n=2 lactacystin B*35:01-YPL treatments.

The online version of this article includes the following source data for figure 7:

**Source data 1.** Monocyte and monocyte-derived dendritic cell cross-presentation with pathway inhibitors.

## Discussion

Our findings, along with previous studies on HLA class I polymorphisms, provide evidence that assembly characteristics associated with various allotypes confer unique advantages and disadvantages for antigen presentation, depending on the cellular environment. In our study, we observe that human monocytes and moDCs differ in their endo-lysosomal pathways, and that HLA-B allotypes vary in their utilization of these pathways. Compared with monocytes, HLA-B has a similar half-life on the surface of moDCs but is less co-localized with LAMP1 (*Figures 2 and 3* and *Figure 3—figure supplement 2*). Furthermore, moDC B*35:01 is peptide-receptive while B*08:01 is not (*Figure 2*), indicating that suboptimal peptide loading of HLA class I is not a global characteristic of moDCs that accounts for shorter moDC half-lives compared to lymphocytes. While previous studies have shown that optimal peptide loading and complex conformation are one set of determinants for cell surface HLA class I stability (*Ljunggren et al., 1990*; *Schumacher et al., 1990*), the present findings suggest that cell-type dependent variations in endocytosis, endocytic maturation, and recycling, could also constitute key determinants of cell surface HLA class I residence time. These studies also place into context our previous findings that relative surface expression levels of individual HLA-B molecules are both cell- and allotype-dependent (*Yarzabek et al., 2018*), a complex product of individual allotype-dependent assembly characteristics and cellular features. Thus, all relative HLA class I expression measurements must be defined in the context of specific cell types.

While it is unclear what specialized factors contribute to the enhanced endocytosis of HLA-Bw6 in moDCs, this is likely a mechanism to promote endo-lysosomal antigen sampling. This hypothesis is validated by the greater co-localization of HLA-Bw6 with Rab11$^+$ endosomes in moDCs compared to other endo-lysosomal markers (*Figure 3*). This differs from monocytes, which have more HLA-Bw6 co-localization with LAMP1$^+$ lysosomes. Greater lysosomal HLA-B localization in monocytes may be a result of enhanced endosome maturation to lysosomes and degradation of endocytosed conformers, as evidenced by the increase in surface HLA-Bw6 upon bafilomycin treatment in *Figure 4*. Monocytes thus appear more poised for lysosomal degradation due to more rapid endosome to lysosome maturation, which is disrupted by bafilomycin treatment (*Bayer et al., 1998*).

In contrast to monocytes, moDCs appear more poised for HLA-B endosomal assembly, particularly for allotypes such as B*35:01 and B*57:01. As Rab11$^+$ endosomes are important storage compartments for MHC class I assembly in APCs (*Montealegre and van Endert, 2018*), the pool of HLA-B present here is indicative of a change in the endosomal system from monocytes for more efficient assembly. Indeed, our data suggest that bafilomycin treatment decreases B*35:01 surface expression because of an inhibitory effect on assembly, as the treatment results both in increased peptide-receptive B*35:01 complexes and greater lysosomal B*35:01 accumulation (*Figure 4*). These findings are the first to our knowledge to demonstrate allotype-dependent differences in constitutive endo-lysosomal assembly of bulk HLA class I proteins. Previous studies have demonstrated MHC class I assembly with endogenous transmembrane proteins (*Tiwari et al., 2007*) and endogenous HSV-1 antigens (*English et al., 2009*) via the endo-lysosomes, and we predict in the context of our findings that presentation of these and other related antigens is allotype-dependent.

Beyond elucidating differences in endogenous HLA-B assembly in the endo-lysosomal system, our findings extend to assembly differences with exogenous antigen (*Figures 6 and 7*). The efficiency of cross-presentation is generally higher for B*35:01 when comparing responses to the same exogenous antigens in the same cells (*Figure 6C–H*), and when comparing across antigens matched for response

sensitivities (*Figure 7C*). Furthermore, monocyte B*35:01-EPL, B*35:01-YPL, and moDC B*35:01-EPL cross-presentation responses all display significant sensitivity to both cathepsin and proteasome inhibition (*Figure 7E*), whereas B*08:01 responses are generally only significantly sensitive to MG132 or lactacystin inhibition (*Figure 7D*). These results suggest that B*35:01 compared to B*08:01 can better exploit vacuolar degradation in monocytes and moDCs to acquire exogenous antigen and thus has greater ability to use multiple (cytosolic and vacuolar) cross-presentation pathways. The locations of processing and assembly may be matched as well; while cathepsins are generally found in lysosomes, cathepsin S has been found located in early endosomal compartments of monocytes and macrophages (*Schmid et al., 2002*). The overall characteristics of B*35:01 that confer the advantages for endo-lysosomal assembly with both endogenous and exogenous antigens include the presence of peptide-receptive forms of B*35:01 to assemble with endosomal-localized antigens (*Figure 8*). Greater efficiency of B*35:01 peptide exchange likely produces more peptide-HLA complexes with EBV peptides that can recycle to the cell surface, increasing the probability of CTLs encountering cognate antigen. Endosomal assembly may also be facilitated by the capability of HLA-B*35:01 to assemble independently of chaperones and factors that are primarily ER localized.

Aside from allotype-dependent differences in cross-presentation efficiencies and pathways, our studies provide important insights into cross-presentation differences between monocytes and moDCs. Whereas resting monocytes are considered non-permissive for cross-presentation (*Döring et al., 2019*), we show their capability for cross-presentation just a few hours after exposure to antigens purified from *Escherichia coli*. Furthermore, the occurrence of more extensive endo-lysosomal antigen degradation in monocytes (*Figure 5*) favors cross-presentation of epitopes with higher affinities, which elicit responses at lower antigen doses (*Figures 6A and 7A*). On the other hand, moDCs are more efficient than monocytes at cross-presentation for most antigens, and antigens with high-response sensitivities appear to be better able to exploit the specialized DC environment to achieve high cross-presentation efficiency (*Figures 6A and 7B*). Both monocytes and moDCs are, to different extents, permissive for both the vacuolar and cytosolic cross-presentation pathways, depending on HLA allotypes and epitopes.

Altogether, based on studies of individual HLA class I allotypes in primary human cells, our findings provide evidence that HLA class I polymorphisms determine not only the specificities of peptide presentation and antigen receptor binding (both innate and adaptive; *Djaoud and Parham, 2020*), but also influence antigen sampling in specific subcellular compartments (*Figure 8*). We suggest that certain HLA class I allotypes are predisposed for bulk constitutive assembly within endo-lysosomes (*Figures 1 and 4*) following suboptimal assembly in the ER, or their peptidome characteristics (*Figure 2*), which enable efficient exchange within endosomal compartments (*Figure 2*). While the sensitivity to bafilomycin partially aligns with previously observed dependencies on PLC components, these assembly characteristics do not fully explain the observed results (*Figure 2D–F*) and imply that a complex mixture of factors dictate dependence on endosomal recycling for maintenance of surface expression. Thus, the subcellular localization of assembly for some HLA class I allotypes in part overlaps with the sites for HLA class II assembly (*Blum et al., 2013*). Whereas in the HLA class II pathway, the invariant chain-derived CLIP peptide maintains an exchange-amenable pool of HLA class II within endo-lysosomal compartments (*Cresswell, 1994*), in the HLA class I pathway, natural variations in the stability and/or peptidome characteristics create a pool of exchange-amenable HLA class I molecules, but in an allotype selective manner. Under steady state conditions, endo-lysosomal HLA class I assembly is likely to be important for maintaining peripheral tolerance against proteins predominantly localized to endo-lysosomes and secreted factors internalized via bulk endocytosis. Allotypes such as B*08:01, with reduced capability for endo-lysosomal assembly, may be more likely to break peripheral tolerance, which could explain some known associations with autoimmune diseases (*Price et al., 1999*; *Candore et al., 2002*; *Miller et al., 2015*; *Rothwell et al., 2016*). Under inflammatory conditions, increased cross-presentation efficiency could lead to increased priming and better activation of antigen-specific CTL responses mediated by allotypes such as B*35:01. Indeed, these findings could explain why, in HIV infections, tapasin-independent allotypes such as B*35:01 have increased breadth of peptide presentation to HIV-specific T cells (*Bashirova et al., 2020*). Additionally, allotypes such as B*35:01 may mediate better protection against pathogens that persist within a sub-cellular endo-lysosomal niche, including *Mycobacterium tuberculosis*, *Toxoplasma gondii,* and *Legionella pneumophila*. More studies are

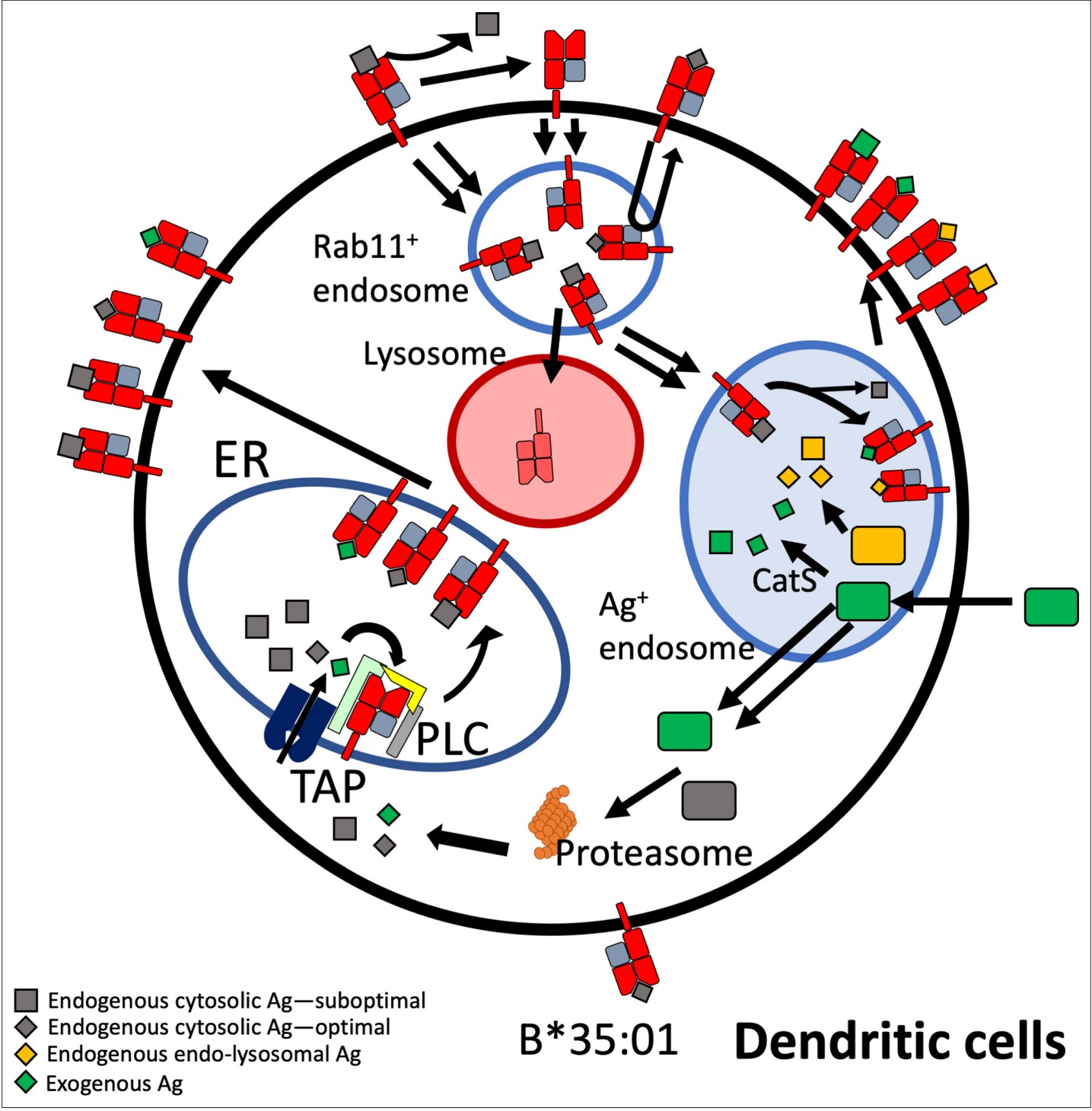

**Figure 8.** Model for B*35:01 endo-lysosomal assembly pathways in monocyte-derived dendritic cells (moDCs). HLA-B*35:01 is assembled in the endoplasmic reticulum (ER), where a greater abundance of suboptimal peptides is predicted to be present due to a mismatch between transporter associated with antigen processing (TAP) specificity and that of B*35:01. The tapasin-independent mode of assembly of B*35:01 could allow for the escape of a fraction of suboptimally loaded B*35:01 from the peptide loading complex (PLC) and ER quality control. Suboptimal B*35:01 complexes on the cell surface are endocytosed. Some B*35:01 molecules are trafficked to Rab11⁺ storage/recycling endosomes or another endosomal compartment where they can be re-assembled with endosomally processed endogenous antigens and recycled to the surface. In a competing pathway, empty or suboptimal conformers are trafficked to lysosomes for degradation. B*35:01 is also recruited from the Rab11⁺ compartment or another endosomal compartment to antigen-containing endosomes, where exogenous antigens are processed in a cathepsin S-dependent manner. B*35:01 can exchange exogenous peptides within these endosomes, followed by recycling back to the cell surface. moDCs can also transport exogenous antigen to the cytosol for proteasomal processing followed by conventional TAP-dependent assembly.

needed to understand the prevalence, extent, and consequences of endo-lysosomal assembly variations among HLA class I allotypes.

## Materials and methods

### Human subjects study approval

Informed consent from healthy donors for blood collections and HLA genotyping was procured in accordance with a University of Michigan IRB approved protocol (HUM00071750). The consent document included information that results of the studies could be published in an article without identifying information about blood donors. Donors were genotyped at the HLA locus as previously described (*Yarzabek et al., 2018*). Alternatively, non-genotyped donor blood was obtained from the University of Michigan Platelet Pharmacology and Physiology core in accordance with a University of Michigan IRB approved protocol (HUM00107120).

### Peripheral blood mononuclear cell isolation

Blood was diluted to 50 mL with PBS + 2% fetal bovine serum (FBS) (PBS/FBS), and 25 mL of diluted blood was overlaid on top of 15 mL Ficoll-paque in two tubes. The tubes were spun at 400 × g for 30 min at RT in a swinging bucket rotor, with the acceleration and deceleration settings set to 4 and 0, respectively. After centrifugation, the top layer of plasma was discarded, and the center layer of cells was collected. PBMCs were washed twice with PBS/FBS for 10 min at 2500 rpm, and PBMCs were resuspended in R10 medium (RPMI + 10% FBS + 1% antibiotic/antimycotic + 1% L-glutamine) and counted.

### Monocyte isolation and moDC generation

For monocyte isolation, cells were purified directly from whole blood or frozen PBMCs using negative magnetic selection. Whole blood was processed using the StemCell EasySep Direct Monocyte Isolation kit (catalog # 19669) for whole blood, or the Miltenyi Classical Monocyte Isolation Kit (catalog # 130-117-337) according to the manufacturer's instructions. After isolation, cells were washed with PBS + 1 mM EDTA (PBS/EDTA), counted, and resuspended in R10 medium at a concentration of 1 million/mL. For moDC differentiation, 6 mL cells (6 million) were added to a well of a six-well plate, and GM-CSF and IL-4 were added to concentrations of 10 ng/mL and 50 ng/mL, respectively. The top 3 mL of medium was replaced with fresh medium + IL-4 and GM-CSF on day 3 and day 5. moDCs were collected for use on day 7.

### HLA-B expression measurements

PBMCs were isolated as described above. The following antibody cocktail was diluted in PBS/FBS and used to stain PBMCs for 30 min on ice to identify various cell populations: anti-CD3-Pacific Blue, anti-CD33-APC/Cy7, anti-CD14-AF700, and anti-HLA-DR-BV650 (all used at 1:200 and from Biolegend). Monocytes were identified as $FSC^{int}$, $SSC^{low}$, $CD3^-$, $CD14^+$, $CD33^+$, and $HLA-DR^+$. For live cell staining and surface HLA-B measurements of PBMCs, cells were aliquoted into a 96-well plate and washed with PBS, followed by staining with 100 μL of antibody cocktail + 1:40 anti-Bw6-FITC or 1:40 anti-Bw4-FITC (OneLambda). For moDC identification after monocyte isolation and differentiation, cells were stained with an antibody cocktail of anti-CD11c-PE/Cy7, anti-HLA-DR-BV650, and anti-CD209-APC (all used at 1:200 and from Biolegend). After staining for 30 min on ice, cells were washed twice with PBS, then stained for 15 min at RT with 7-AAD (1:200, BD), and followed by analysis on a flow cytometer. For inhibitor treatments, PBMCs or moDCs were treated in a 96-well plate with either bafilomycin A1 (200 nM, Cayman Chemical catalog #11038) or MG132 (10 μg/mL, Sigma catalog # 474787) for various time points, and the above staining protocol was followed.

   For surface and total HLA-B or HLA-C measurements of moDCs, cells were first stained with Red Fixable Live/Dead dye diluted in PBS for 15 min at RT (1:1000, ThermoFisher). Cells were washed with PBS, followed by fixation with 4% PFA diluted in PBS for 10 min at RT. Next, cells were washed with PBS, and half of the samples stained with the antibody cocktail + anti-Bw6 (surface HLA-Bw6), and half stained with the antibody cocktail alone. Cells were stained for 30 min on ice then washed twice. The cells stained for surface HLA-Bw6 were set aside, and the remaining cells were stained with anti-Bw6

diluted in 0.2% saponin (total HLA-Bw6) for 30 min on ice. After washing twice, both sets of cells were analyzed on a BD LSR Fortessa flow cytometer. Data was analyzed using FlowJo software.

## moDC HLA-B half-life measurements

Surface stability and half-life assessment were performed as described previously (*Zarling et al., 2003*; *Yarzabek et al., 2018*). Briefly, monocytes were isolated and differentiated to moDCs for 7 d as described above. B*08:01+ or B*35:01+ donor moDCs were plated into a 96-well plate in duplicate for each condition. BFA treatment was added at negative time points: for a 4-hr time course, BFA was first added to cells for the 4-hr treatment time point, then 3 hr, etc. BFA was added at a concentration of 0.5 μg/mL to each well in media, and cells were incubated before centrifugation, washing with PBS, and staining with a monoclonal antibody cocktail of anti-CD11c-PE/Cy7, anti-HLA-DR-BV650, and anti-CD209-APC (all used at 1:200 and from Biolegend), as well as anti-Bw6-FITC (Biolegend, 1:40), for 30 min on ice. After staining, cells were washed twice with PBS, followed by staining with 7-AAD and analysis on a BD LSR Fortessa flow cytometer. Half-life values were extracted using a one-phase decay curve with a constrained plateau of zero.

## moDC HLA-B thermal stability

Surface HLA-B thermal stability was assessed based on previously established methods. In brief, moDCs were incubated for either 1 or 2 hr at RT, 37 or 42°C. Following incubation, moDCs were washed and stained with a monoclonal antibody cocktail of anti-CD11c-PE/Cy7, anti-HLA-DR-BV650, and anti-CD209-APC (all used at 1:200 and from Biolegend), as well as anti-Bw6-FITC (Biolegend, 1:40), for 30 min on ice. After staining, cells were washed twice with PBS, followed by staining with Red live/dead fixable dye and analysis on a BD LSR Fortessa flow cytometer. Expression values for each allotype were normalized to 37°C, and each allotype was compared using unpaired t tests.

## HLA-B bead-based pH peptide exchange assay

HLA-B*08:01 or HLA-B*35:01 monomers were acquired from the NIH Tetramer Core. Biotinylated monomers were digested with PreScission Protease (PsP) at 0.1 mg/mL overnight at RT, pH 7.0 to remove the covalently linked peptide from the peptide-binding groove. In addition to PsP, low-affinity peptides (DANDIYRIF for B*08:01 or APLLRWVL for B*35:01) were added at 100 μM. Following monomer digestion and loading, monomers were washed three times with pH 7 MES buffer to remove excess peptide, followed by incubation at 37°C for 1.5 hr with no peptide, medium-affinity peptides, or high-affinity peptides. For B*08:01, these peptides were: GPKVKRPPI, RAKFKQLL, or FLRGRAYGL. For B*35:01, these peptides were: HPVGEADYFEY or EPLPQGQLTAY. For peptide exchange, monomers were diluted to 5.55 μg/mL (~1 μM), and peptide was added to 90.6 μM. Exchange was carried out in MES buffer of pH 4, 5, 6, or 7. Following this incubation, 100,000 5 μm streptavidin-coated microspheres were added to each reaction to bind the monomers, and the reactions were returned to 37°C for 1 hr. Following peptide binding and coating to microspheres, the monomer/microsphere complexes were pelleted at 4500 rpm for 20 min at 4°C. The reactions were washed with PBS + 2 mM EDTA + 2% FBS. In 50 μL of PBS/EDTA/FBS buffer, monomers bound to microspheres were stained with the monoclonal antibodies HC10-FITC or BBM.1-FITC (both prepared from ascites and labeled in-house). Antibody staining was performed on ice for 30 min, followed by washing with PBS/EDTA/FBS and analysis on a BD LSR Fortessa flow cytometer. HC10 or BBM.1 signals of B*08:01 or B*35:01 monomers loaded with low-affinity peptide followed by no peptide addition were compared to those loaded with medium-affinity or high-affinity peptide to quantify the peptide exchange at each pH.

## moDC peptide receptivity

Peptides were synthesized by A&A Labs LLC or Peptide 2.0. Monocytes were isolated and differentiated to moDCs for 7 d as described above. B*08:01+ or B*35:01+ donor moDCs were plated into a 96-well plate in duplicate for each condition, followed by the addition of either DMSO, canonical peptide (100 μM), or control peptide (100 μM). Control peptides were truncated and altered at anchor residues relative to the canonical peptide sequence. For B*08:01, the canonical peptides used were GPKVKRPPI, RAKFKQLL, or FLRGRAYGL, and the control peptides used were GPDVERPP, RADFEQLG, or FLDGEAYGG. For B*35:01, the canonical peptides were HPVGEADYFEY or EPLPQGQLTAY, and the control peptides used were HGVGEADYFE or EGLPQGQLTA. Peptides were

incubated with moDCs for 4 hr at 37°C, followed by washing and staining with antibodies for moDC surface markers, as well as the monoclonal antibody HC10-FITC, which recognizes peptide-deficient conformers of HLA class I, or the monoclonal antibody BBM.1-FITC, which recognizes $\beta_2$m molecules in complex with HLA class I. After 30 min of staining on ice, cells were washed twice with PBS, stained with 7-AAD (1:200), and analyzed on a BD LSR Fortessa flow cytometer. Additional experiments were performed where peptide incubation was performed at 4°C, or at 37°C in the presence of 200 nM bafilomycin. Data was analyzed using FlowJo software.

## Confocal microscopy

Monocytes were isolated from blood as described above. Glass coverslips were coated with poly-L-lysine for 2 hr at 37°C in 12-well plates, then washed 3× with water and allowed to dry completely. For each coverslip, about 250,000 monocytes were added in 100 µL medium and allowed to adhere to the coverslips for 2 hr at 37°C. Coverslips were washed with PBS gently, then fixed with 4% PFA for 10 min at RT. Coverslips were washed with PBS, then permeabilized with 0.1% Triton X-100 in PBS for 10 min at RT. Coverslips were washed twice with PBS, then blocked with 5% goat serum diluted in PBS + 0.05% Tween 20 (PBST). Blocking was performed for 1 hr at RT with gentle rocking. After blocking, coverslips were inverted onto a 100-µL bubble of primary antibody staining solution (diluted antibody + 1% BSA in PBST) placed on a piece of parafilm and incubated in a cold room overnight. The next day, the coverslips were returned to a 12-well plate and washed 3× with PBST for 5 min each with rocking. Coverslips were stained with 500 µL secondary antibody solution (antibody diluted in PBST + 1% BSA) for 1 hr at RT while rocking. Coverslips were washed 3× with PBST for 5 min each with rocking and inverted onto a 15 µL drop of ProLong Diamond + DAPI placed on a glass slide. Slides were cured at RT overnight, then sealed with nail polish around the edges. Images were acquired with a Nikon A1 confocal microscope using a pinhole size of 1, a z-step of 0.3 um, a pixel dwell time of 12.1, and a line average of 2. Primary antibodies used were: mouse anti-Bw6-biotin (1:20, OneLambda), rabbit anti-EEA1 (1:1000, Invitrogen), mouse IgG1 anti-Arf6 (1:20, Invitrogen), rabbit anti-Rab11a (1:12.5, ThermoFisher), and rabbit anti-LAMP1 (1:100, CellSignaling Technologies). Secondary antibodies/probes used were: streptavidin-AF488 (1:2000, Invitrogen), streptavidin-AF647 (1:2000, Invitrogen), goat anti-rabbit-AF555 (1:500, Abcam), and goat anti-mouse IgG1-AF488 (1:500, Invitrogen).

For confocal experiments using bafilomycin treatment, moDCs were cultured and allowed to adhere to coverslips as described above. After adherence for 1–2 hr, culture media in each well was replaced with either 1 mL R10 for the untreated controls or 1 mL R10 + 200 nM bafilomycin for treatment conditions. Cells were cultured with inhibitor for 4 hr, followed by washing with PBS, fixation for 10 min with 4% PFA, and staining for Bw6/Rab11a or Bw6/LAMP1 as described above.

Co-localization was assessed using one of two methods. For object-based co-localization, a FIJI macro was written based on a previously described method (*Moser et al., 2017*). Briefly, image files were analyzed using an identical macro script involving background masking and subtraction, signal thresholding, and quantification of the fraction of signal A that overlaps spatially with signal B. Pearson's correlation was performed uniformly to each image file using the JACOP plugin for FIJI (*Bolte and Cordelières, 2006*), which quantifies the correlation of co-occurrence of bright pixels of signal A and bright pixels of signal B.

## Antigen uptake time course

Monocytes or moDCs were plated into a 96-well plate at 100,000 cells/well. Cells were pulsed with 10 µg/mL BSA labeled in-house with Alexa Fluor 594 in duplicate for either 15 min, 30 min, 45 min, or 60 min. Duplicate wells were left untreated, and duplicate wells were pulsed with BSA-AF594 for 60 min on ice to inhibit endocytosis and measure background fluorescence. After each time point, cells were collected and washed twice with PBS, followed by fixation with 4% PFA for 5 min at RT. Cells were washed and analyzed by flow cytometry using a BD LSR Fortessa. Data was analyzed using FlowJo software.

## DQ-Ova antigen processing assays

Monocytes or moDCs were plated into a 96-well plate at 100,000 cells/well. Cells were pulsed with 50 µg/mL DQ-Ova antigen in media for 30 min at either 37 or 4 °C. Following 30 min pulse, cells were washed with PBS and chased in either media alone, media + 200 nM bafilomycin, media + 10 µg/mL

MG132, or media + bafilomycin and MG132 for 2 hr (monocytes) or 4 hr (moDCs). Control cells which were pulsed with DQ-Ova at 4°C were also chased in media at 4°C for 2 or 4 hr. Following chase, cells were washed with PBS and stained with Aqua fixable live/dead (1:500 in PBS, ThermoFisher catalog #L34957) for 15 min at RT. Cells were washed again with PBS and fixed with 4% PFA for 10 min at RT. Samples were measured by flow cytometry using a BD LSR Fortessa, and analyzed using FlowJo software.

## Polyclonal antigen-specific CTL expansion

CD8+ T cells were isolated by negative magnetic selection from either B*08:01+ or B*35:01+ donor blood using kits from StemCell Technologies (catalog #17953). To screen for antigen-specific CTLs, common EBV epitopes for a particular HLA-B allele were identified from the Immune Epitope Database (IEDB) and used to produce peptides. These peptides were loaded onto B*08:01 or B*35:01 monomers as described above, and peptide-loaded monomers were bound to streptavidin-APC molecules for flow cytometric staining. Isolated CD8+ T cells were stained with tetramer (1:20) for 1 hr on ice. Cells were washed with PBS and stained with anti-CD3-Pacific Blue (1:200, Biolegend catalog #300417) and anti-CD8-AF700 (1:200, Biolegend catalog #344724) antibodies diluted in PBS/FBS for 30 min on ice. Cells were washed twice and stained with 7-AAD (1:200) and analyzed by flow cytometry. Tetramer positive cells were identified and sorted via fluorescence-activated cell sorting.

Sorted tetramer-specific CTLs were expanded as previously described (*Dong et al., 2010*). Briefly, HLA class I allo PBMCs were isolated and resuspended to a concentration of 2 million cells/mL. Cells were irradiated at 3300 rad (performed at the UMich Experimental Irradiation Core) and plated at 200,000 cells/well in a 96-well plate with 3.2 µg/mL phytohemagglutinin (PHA, Remel catalog # R30852801). About 1000 sorted cells were added to each well, to a final volume of 200 µL/well. Twice per week, the top 100 µL of media was removed from each well, and fresh media + 10 µL natural human IL-2 (Hemagen Diagnostics, catalog # 906011) was added. CTLs were maintained in this feeder cell expansion culture for 2–3 wk, checking the expansion of tetramer+ cells by the end with of the expansion. Following expansion in 96-well plate format, CTLs were transferred to a T-25 flask with 25–50 million irradiated allo-PBMCs, 1 µg/mL PHA, and 50 U/mL recombinant human IL-2 (Peprotech, catalog # 200–02) in 25 mL media. Expansion in flasks was continued for another 2–3 wk, checking with tetramer staining periodically. CTL density was kept under 2 million cells/mL, splitting into a new flask as needed during the expansion. After expansion, CTLs were frozen down into aliquots of about 5 million cells per cryovial. For use in activation assays, a CTL vial was thawed and added to a T-25 flask with 25–50 million irradiated allo-mismatched PBMCs, PHA, and IL-2. CTLs typically re-expanded to a usable density in about 1–1.5 wk and were ready to use for activation assays for the next 1–2 mo.

## CTL peptide titration

PBMCs from B*08:01+ or B*35:01+ were pulsed overnight with media alone, or media+ serially diluted peptides. Peptide concentrations ranged from 1 pM to 100 µM, with tenfold dilutions. PBMCs pulsed in duplicate for each condition, at 100,000 cells/well in a 96-well plate. After overnight incubation with peptide, PBMCs were washed and co-cultured with CTLs specific for each peptide at a 2:1 CTL:PBMC ratio (200,000 CTLs/well). GolgiStop and GolgiPlug inhibitors (1:800) were included in the co-culture to block cytokine export, and an anti-CD107a-PE antibody (1:200) was added to stain for CTL degranulation. After a 5-hr co-culture, cells were washed and stained with Red Fixable Live/Dead dye (1:1000) for 15 min at RT. Cells were washed, then stained with anti-CD3-Pacific Blue (1:200) and anti-CD8-AF700 (1:200) antibodies diluted in PBS/FBS for 30 min on ice. Cells were washed twice and fixed for 10 min at RT with 4% PFA diluted in PBS. After fixation, cells were stained intracellularly with anti-IFNγ-FITC (1:100, Biolegend catalog #506504) diluted in 0.2% saponin/PBS for 30 min on ice. Cells were washed twice and analyzed by flow cytometry using a BD LSR Fortessa. Data was analyzed by FlowJo.

## Plasmid cloning

A BZLF1 encoding sequence was synthesized by Integrated DNA Technologies (IDT) based on the European Nucleotide Archive coding sequence (accession # AAA66529). Primers were used to perform PCR to amplify the sequence:

5′-TAAGCAGGATCCATGATGGACCCAAACTCGAC-3′ and 5′-TGCTTAGCGGCCGCTTAGAAA TTTAAGAGATCCT-3′.

The primers inserted a BamHI site upstream of the BZLF1 gene, and a NotI site downstream of the gene, with about six nucleotides of overhang on either side of the PCR product. PCR product was digested with BamHI and NotI, after which the enzymes were removed using a PCR Cleanup Kit (Qiagen). The vector pGEX-4T-LP was digested using the same enzymes, ran on a 1% agarose gel, excised, and cleaned up using a Gel Extraction Kit (Qiagen). The BZLF1 PCR product was ligated into the pGEX vector using T4 DNA ligase at 16°C overnight. The ligation product was transformed into Rosetta cells, which were selected on LB-Ampicillin plates. Colonies were picked, plasmid DNA isolated, and sequenced to confirm gene insertion. The BamHI site places the BZLF1 gene in-frame downstream of the GST protein, creating a GST-BZLF1 fusion protein containing a thrombin cleavage site between the proteins.

The EBNA3A:133–491 (simplified to EBNA3A throughout) truncation protein was cloned by PCR amplification of a segment of the gene encoding residues 133–491. This segment contained the coding sequence for both the FLRGRAYGL and YPLHEQHGM epitopes. The MSCV-N EBNA3A plasmid used as a backbone for PCR was a gift from Karl Munger (Addgene plasmid # 37956; http:// n2t.net/addgene:37956.; RRID:Addgene_37956) (*Rozenblatt-Rosen et al., 2012*). Primers used to amplify the EBNA3A:133–491 gene segment:

5′- CGCGTGGATCCATGTACATAATGTATGCCATGGC –3′ and 5′- ACGATGCGGCCGCTTAAACA CCTGGGAGTTG -3′.

The primers inserted a BamHI site upstream of the EBNA3A:133–491 gene, and a NotI site down-stream of the gene, with about six nucleotides of overhang on either side of the PCR product. Cloning proceeded as described for BZLF1 above.

## Protein expression and purification

GST-BZLF1 protein or GST-EBNA3A:133–491 protein was purified based on the protocol by *Harper and Speicher, 2011*. Briefly, Rosetta cells were transformed with either BZLF1-pGEX-4T-LP or EBNA3A-pGEX-4T-LP plasmids and plated on LB-Amp plates, and single colonies were picked and used to inoculate a 30-mL LB-Amp overnight starter culture. The next day, 10 mL of culture was used to inoculate a 1 L flask of LB-Amp. The flask was grown shaking at 37°C until the culture reached an OD600 of between 0.5 and 0.7. The flask was then cooled to 16°C, and IPTG was added to a final concentration of 1 mM. Protein expression was induced overnight at 16°C with shaking. The next day, the culture was split in half and spun down for 30 min at 4000 g at 4°C. One pellet was stored at –80°C, and the other was resuspended in 15 mL lysis buffer (50 mM Tris + 1% Triton X-100 + 5 mM EDTA + 1 mM 2-ME+0.15 mM PMSF + 1 cOmplete protease inhibitor tablet/50 mL, pH = 8.0) and lysed by sonication. The lysate was spun at 13,000 rpm at 4°C for 30 min, and the supernatant was applied to glutathione resin. The supernatant and column were incubated gently rocking at 4°C for 2 hr. The column was thoroughly washed with 100 mL PBS + 5 mM EDTA + 0.15 mM PMSF (PBS/ EDTA/PMSF), followed by 100 mL PBS/EDTA. 20 mL of 20 mM reduced L-glutathione was applied to the column and allowed to sit at RT for 30 min. The glutathione solution was allowed to slowly move through the column, and eluted protein was collected in 1 mL fractions. Eluted fractions were quantified for protein concentration using a nanodrop, and selected fractions were analyzed using 12% SDS PAGE gel alongside lysate fractions to check protein purity and size. Fractions containing the GST-BZLF1 or GST-EBNA3A fusion were concentrated using a 10-k molecular weight cutoff centricon and buffer exchanged into PBS buffer. Protein was concentrated typically to a concentration of about 2 mg/mL and stored at –20°C.

## Cross-presentation assays

Monocytes or moDCs were plated into sterile 96-well plates at 50,000–100,000 cells/well. In quadruplicate, APCs were pulsed with either no antigen, peptide (50 μM), 100 μg GST-BZLF1, or 100 μg GST-EBNA3A. For the inhibitor experiments, cells were pulsed with 100 μg protein (GST-BZLF1 or GST-EBNA3A), 100 μg protein + 50 μM CTI1 (Selleck Chem catalog # S2847), 100 μg protein + 10 μg/mL MG132 (Sigma catalog # 474787), or 100 μg protein + 10 μM lactacystin (Sigma catalog # L6785). Antigen pulses with and without inhibitors were performed for 6 hr at 37°C, followed by washing with PBS. The two antigen-specific CTLs used for comparison were

added at a 1:1 CTL:APC ratio so that each CTL was added to each condition in duplicate. Also added to the culture with the CTLs was anti-CD107a-PE (1:20, BD, or 1:200, Biolegend), GolgiStop (1:800, BD), and GolgiPlug (1:800, BD). Cells were spun for 3 min at 1200 rpm and co-incubated for 5 hr. After co-culture, cells were washed and stained with Red Fixable Live/Dead dye (1:1000, ThermoFisher catalog #L34971) for 15 min at RT. Cells were washed, then stained with anti-CD3-Pacific Blue (1:200, Biolegend catalog #300417) and anti-CD8-AF700 (1:200, Biolegend catalog #344724) antibodies diluted in PBS/FBS for 30 min on ice. Cells were washed twice and fixed for 10 min at RT with 4% PFA diluted in PBS. After fixation, cells were stained intracellularly with anti-IFNγ-FITC (1:100, Biolegend catalog #506504) diluted in 0.2% saponin/PBS for 30 min on ice. Cells were washed twice and analyzed by flow cytometry using a BD LSR Fortessa. Data was analyzed by FlowJo.

## Materials availability statement

Materials, with the exception of primary cells, can be procured by writing to the corresponding author.

## Acknowledgements

We thank all of our study participants for their time and participation in the study, as well as the Michigan Clinical Research Unit (MCRU) and Michigan Medicine blood drawn station staff for performing blood draws. We thank Dr. Anita Zaitouna for the Shannon Entropy analyses for *Figure 1F*, Kole Tison for assistance with assay development for *Figure 2D*, and the laboratory of Dr. David Koelle for providing detailed protocols for the isolation and expansion of antigen specific CD8[+] T cells. We thank the NIH Tetramer Core for the B*08:01 and B*35:01 proteins used for the isolation of antigen specific CD8[+] T cells and for characterizing the effects of peptides on HC10 and BBM1 binding. We thank Aaron Taylor and Eric Rentchler at the Microscopy Core of the University of Michigan Biomedical Research Core Facility and Miles Mckenna (Nikon Instruments Inc) for assistance with user training or microscopy image analysis. We thank Amanda Prieur and the UM Platelet Core for providing non-genotyped blood for experiments. We also thank staff at the Flow Cytometry Core the University of Michigan Biomedical Research Core Facility for assistance with cell sorting and Dr. David Karnak and the Experimental Irradiation Core at Michigan for assistance producing irradiated feeder cells for CTL expansion. We are grateful to Drs. Irina Grigorova, Joel Swanson, Kathleen Collins, and Pavan Reddy for helpful scientific input. This work was supported by NIH grant RO1 AI044115 and R21 AI64025 (to MR), NIH training grants T32GM008353 and T32AI007413 (to EO), and the Herman and Dorothy Miller Fund (to EO).

## Additional information

### Funding

| Funder | Grant reference number | Author |
|---|---|---|
| National Institute of Allergy and Infectious Diseases | RO1AI044115 | Malini Raghavan |
| National Institute of Allergy and Infectious Diseases | R21AI64025 | Malini Raghavan |
| National Institute of General Medical Sciences | T32GM008353 | Eli Olson |
| National Institute of Allergy and Infectious Diseases | T32AI007413 | Eli Olson |
| Herman and Dorothy Miller Fund | | Eli Olson |

The funders had no role in study design, data collection and interpretation, or the decision to submit the work for publication.

### Author contributions
Eli Olson, Formal analysis, Validation, Investigation, Visualization, Methodology, Writing - original draft, Writing – review and editing; Theadora Ceccarelli, Validation, Investigation, Visualization, Methodology; Malini Raghavan, Conceptualization, Formal analysis, Supervision, Funding acquisition, Visualization, Project administration, Writing – review and editing

### Author ORCIDs
Eli Olson [ID] http://orcid.org/0000-0003-2319-7144
Theadora Ceccarelli [ID] http://orcid.org/0000-0002-5010-1008
Malini Raghavan [ID] http://orcid.org/0000-0002-1345-9318

### Ethics
Human subjects: Informed consent from healthy donors for blood collections and HLA genotyping was procured in accordance with a University of Michigan IRB approved protocol (HUM00071750). The consent document included information that results of the studies could be published in an article without identifying information about blood donors. Donors were genotyped at the HLA locus as previously described (Yarzabek et al., 2018). Alternatively, non-genotyped donor blood was obtained from the University of Michigan Platelet Pharmacology and Physiology core in accordance with a University of Michigan IRB approved protocol (HUM00107120).

### Decision letter and Author response
Decision letter https://doi.org/10.7554/eLife.79144.sa1
Author response https://doi.org/10.7554/eLife.79144.sa2

## Additional files

### Supplementary files
• Supplementary file 1. Healthy human donors and human leukocyte antigen (HLA) genotypes used in study. Donors were selected from our previously described cohort of HLA genotyped healthy participants (*Yarzabek et al., 2018*). Ten primary groups were recruited for this study. Groups 1–8 were selected to be either homozygous for an HLA-B allotype of interest, or heterozygous for one Bw6 allotype and one Bw4 allotype so that anti-Bw6 and anti-Bw4 monoclonal antibodies could be utilized to measure the specific expression/localization of individual HLA-B allotypes. Donors in these groups were selected to have one or less HLA-C allotype that cross-reacts with the anti-Bw6 antibody, or no HLA-A allotypes that cross-react with the anti-Bw4 antibody (*Yarzabek et al., 2018*). Two cross-reactive HLA-C allotypes (denoted by asterisks) are permitted if the donor is homozygous for the HLA-B allotype of intertest, or if assays not using anti-Bw6 were not performed (e.g. peptide receptivity assays with the HC10 monoclonal antibody). Group 9 donors expressed B*08:01, B*35:01, or both and were selected for use as effector antigen-specific CTLs for all T cell activation assays. Heterozygosity for Bw6/Bw4 was not required for studies with this group. Group 10 contained donors heterozygous for B*08:01 and B*35:01 and were used as antigen-presenting cells for cross-presentation assays. For assays where specific HLA genotype was not required, such as DQ-Ova experiments, non-genotyped donors 248–275 were used, or blood was obtained from the University of Michigan Platelet Core. Donors from the Platelet Core are labeled.

• MDAR checklist

### Data availability
The original data have been deposited to Dryad. Source data for figures have also been provided.

The following dataset was generated:

| Author(s) | Year | Dataset title | Dataset URL | Database and Identifier |
|---|---|---|---|---|
| Raghavan M, Olson E, Ceccarelli T | 2023 | Endo-lysosomal assembly variations among Human Leukocyte Antigen class I (HLA-I) allotypes | https://doi.org/10.5061/dryad.qbzkh18nb | Dryad Digital Repository, 10.5061/dryad.qbzkh18nb |

The following previously published datasets were used:

| Author(s) | Year | Dataset title | Dataset URL | Database and Identifier |
|-----------|------|---------------|-------------|-------------------------|
| Sarkizova et al | 2020 | HLA-I peptidomes | ftp://massive.ucsd.edu/MSV000084172/ | MassIVE, MSV000084172 |
| Sarkizova et al | 2020 | HLA-I peptidomes | ftp://massive.ucsd.edu/MSV000080527 | MassIVE, MSV000080527 |

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
