## [Editor Report]

This study provides novel insights into the role of HLA polymorphisms in the processing of exogenous antigens. The evidence supporting the conclusions is strong, with rigorous cellular and biochemical assays. The work will be of interest to scientists within the field of antigen presentation.

---

## [Decision Letter]

**Decision letter after peer review:**

Thank you for submitting your article "Endo-lysosomal assembly variations among Human Leukocyte Antigen (HLA) class I allotypes" for consideration by *eLife*. Your article has been reviewed by 3 peer reviewers, and the evaluation has been overseen by a Reviewing Editor and Tadatsugu Taniguchi as the Senior Editor. The reviewers have opted to remain anonymous.

Essential revisions:

The manuscript is conceptually innovative and of fundamental interest to the classical MHC-I antigen presentation and cross-presentation field. However, additional experiments are required to strengthen the conclusions of this manuscript.

*Reviewer #1 (Recommendations for the authors):*

In general, the work is thoughtfully done but is restricted to qualitative serologic assays. The work can be bolstered in a few ways to make quantitative arguments.

Key qualitative serologic assays should be complemented with quantitative cellular biochemical assays. For example, in a heterologous system using H-2KbDb-deficient mouse BM-derived DCs, express single B8.1 or B35.1 independently and biochemically characterize intracellular trafficking, half-life, peptide receptivity, et cetera. In such a system one could take advantage of mAb-tags (FLAG, myc, HA, andC) to precisely monitor heavy chain synthesis and assembly with a combination of conformation-independent HC10/tag-specific mAbs and conformation-dependent Bw6-specific mAbs.

Biochemically quantify peptide receptivity: This is a standard assay established in the 90s by Pete Cresswell, Hidde Ploegh, Jon Yewdell, and others for MHC-I molecules. When performed at 4, 15, 26, 37 and 42 oC, one can get a quantitative idea of HLA-I stability and peptide receptivity.

See paper by Cresswell (DOI: 10.1007/BF00211552); according to this early study of B7 trafficking in T2 cells shows TAP dependency. The same would be true of most HLA-I but for those that bind with signal peptides or peptides that enter the ER lumen by other mechanisms than TAP transport. In this regard, it would be interesting to test if forced overexpression of at least one B35.1-restricted peptide via ER-targeted mini-peptide construct would rescue ER assembly of B35.1.

The cellular site of peptide rescue of HLA-I is hard to interpret without additional experimentation. See www.doi.org/10.1016/s1074-7613(95)80014-x.

The co-localization micrographs are very small, and the resolution is low. These need to be bolstered.

Can engineering tapasin binding site/s into B35.1 rescue ER assembly of stable B35.1? In this regard, it is curious nothing is said or tested about the role of TAPBPR-a tapasin homologue that functions outside of the PLC in MHC-I/HLA-I. assembly of stable complexes.

MG132 is a reasonable proteasome inhibitor but it has limited specificity. Critical experiments should be repeated with lactacystin, which is a gold standard for proteasome inhibition; an alternative is epoxomicin, an irreversible proteasome inhibitor.

It is quite evident that the two recombinant EBV-derived proteins are cross presented to respective T cells. The evidence in this report suggests that the cytoplasmic pathway dominates the endo-lysosomal pathway in cross-presentation. Using a broader spectrum proteasome inhibitor may help amplify the role of proteasomes and, hence, the cytoplasmic pathway to cross-presentation.

Now, there is some evidence that autophagy can dump proteasomes into lysosomes. Considering that cross-presenting DCs are constitutively autophagic, which have roles in antigen cross-presentation by mouse MHC-I, are the outcomes of MG132 inhibition a result of proteasome activity in these late vesicles?

Lastly, this work is narrowly focused on a single member of the B7 supertype-B*35:01, which is prevalent at low frequency in the African American (AA: 0.05) and White American (WA: 0.07) populations. Extension of current work beyond B35.1 and the Bw6 serotype by more quantitative cellular biochemical approaches to one or more additional allotypes in the B7 superfamily-e.g., B7.2 (AA: 0.08; WA: 0.155) itself and/or B35.3 (AA: 0.005; WA: 0.027) or B53.1 (AA: 0.133; WA: 0.004), could bolster and reveal to the importance of the findings in this report. Outcomes may also yield insights into the causes and maintenance of HLA-I polymorphisms.

*Reviewer #3 (Recommendations for the authors):*

1) The introduction of the manuscript is rather long (5 paragraphs, 3.5 pages) for a research article. While this provides important details, it may also detract from the presentation of the results. Consider shortening and consolidating paragraphs in the introduction.

2) P8, pp 2: Regarding the very elegant assay using HC10 can probe empty conformers: What are the background levels arising from staining of peptide-loaded molecules using this antibody for the two B allotypes, under the conditions of the experiments? One way to account for this is to incubate with either Bw6 or W6/32 first as blocking antibodies, and then measure background levels with HC10.

3) P11, pp3 and Figure 3G: HC10 suggests a reduction in B35 (but not B08) peptide receptivity upon bafilomycin treatment, measured by HC10 staining ratio. Are the receptive molecules properly conformed for peptide binding, and do they contain b2m? This is an important detail since the loss of b2m will leads to the formation of heavy chain dimers and re-internalization of MHC-I molecules (see, for example, Dirscherl et al., J cell Sci, 2022).

4) P13, pp1: the authors employ assays using (likely oligoclonal) tetramer-sorted T cells to probe the formation of properly conformed peptide:MHC-I complexes on the cell surface, however, there could exist cross-reactivity with different antigens which may convolute the results of this analysis. Please provide evidence for the lack of cross-reactivity for the different EBV epitopes arising from the same protein, using tetramer staining or a similar assay to complement the analysis.

5) P 14, pp3: the authors provide evidence of different response sensitivities for different EBV epitopes, depending on contributions from the cytosolic or vacuolar pathways. Can they elaborate further on plausible mechanisms for the observed differences, for example, changes in surface expression levels, densities, or distribution of molecules, and if possible, provide a quantitative assessment of the number of pMHC molecules under different conditions?

6) Are there differences in glycosylation of MHC molecules at the conserved Asn 86 position processed through the vacuolar or cytosolic pathways? If so, it would be interesting to see if the contribution of different pathways, such as in the case of B08 vs B35 affects the complex glycosylation patterns on the cell surface.

7) Figure 7 (the summary figure) is too complex, with too many graphical elements. For a conceptual model figure, this may confuse readers. Consider simplifying by reducing the number of pMHC complexes illustrated.

---

## [Author Response]

Reviewer #1 (Recommendations for the authors):In general, the work is thoughtfully done but is restricted to qualitative serologic assays. The work can be bolstered in a few ways to make quantitative arguments.Key qualitative serologic assays should be complemented with quantitative cellular biochemical assays. For example, in a heterologous system using H-2KbDb-deficient mouse BM-derived DCs, express single B8.1 or B35.1 independently and biochemically characterize intracellular trafficking, half-life, peptide receptivity, et cetera. In such a system one could take advantage of mAb-tags (FLAG, myc, HA, andC) to precisely monitor heavy chain synthesis and assembly with a combination of conformation-independent HC10/tag-specific mAbs and conformation-dependent Bw6-specific mAbs.

Thank you for this suggestion. While an analysis of epitope-tagged versions of HLA-B allotypes would certainly be advantageous for some assays, we sought to retain endogenous HLA-B expression conditions and human peptide loading complex (PLC) components in primary human cells in order to preserve both the biologically relevant ratios of HLA-I:PLC, and optimal PLC functionality. Our concerns were that single HLA-B expression in BM-DCs would result in altered HLA-:PLC stoichiometries compared with human moDCs. Additionally, there are known functional differences between murine and human tapasin (1-3) and between HLA- heavy chain complexes with human vs mouse β_2_m (4), which could differentially influence cellular HLA-B assembly characteristics in BM-DCs. We want to emphasize that the use of anti-Bw6 allows the measurement of single HLA-B allotypes, based on the selection of Bw6/Bw4 heterozygous donors. Based on the much lower HLA-C expression (Figure 1—supplement 1), the expression and stability measurements are specific for the indicated HLA-B. This point has been thoroughly addressed in a previous *eLife* paper (5).

Biochemically quantify peptide receptivity: This is a standard assay established in the 90s by Pete Cresswell, Hidde Ploegh, Jon Yewdell, and others for MHC-I molecules. When performed at 4, 15, 26, 37 and 42 oC, one can get a quantitative idea of HLA-I stability and peptide receptivity.

We undertook new experiments to measure and compare the thermal stabilities of surface HLA-B. These are shown in a new Figure 2C. Overall, Bw6 expression at 42°C was lower than at 37°C, and significant differences are not measured between the two HLA-B allotypes. We believe that the half-life calculations of Figures 2A and 2B are more sensitive (and more quantitative). The measured stability differences between B*08:01 and B*35:01 in primary cells expressing a normal repertoire of PLC proteins (Figures 2A and 2B) (and Yarzabek et al 2018) may be more subtle than differences that have been measured between various allotypes in cell lines deficient in specific PLC components. These points are clarified on page 7.

See paper by Cresswell (DOI: 10.1007/BF00211552); according to this early study of B7 trafficking in T2 cells shows TAP dependency. The same would be true of most HLA-I but for those that bind with signal peptides or peptides that enter the ER lumen by other mechanisms than TAP transport. In this regard, it would be interesting to test if forced overexpression of at least one B35.1-restricted peptide via ER-targeted mini-peptide construct would rescue ER assembly of B35.1.

We think this is a highly interesting suggestion, but believe it would be better suited as an independent follow-up study, including how expression of the ER-targeted peptide would affect the bafilomycin sensitivity and cross-presentation characteristics of B*35:01.

The cellular site of peptide rescue of HLA-I is hard to interpret without additional experimentation. See www.doi.org/10.1016/s1074-7613(95)80014-x.

We thank the reviewer for pointing this out, and agree that the site of peptide binding during the peptide receptivity assays is unclear. We have added a section acknowledging this possibility on Page 8 of the manuscript. Our main focus with the receptivity assays was to demonstrate that B*35:01 is suboptimally assembled to a greater extent than B*08:01 and more amenable to binding exogenous peptide.

The co-localization micrographs are very small, and the resolution is low. These need to be bolstered.

The current images were obtained from specific HLA genotyped blood donors over a period of 3 years using a Nikon confocal microscope. The microscopy was done in consultation with the Microscopy Core staff at the University of Michigan on the use of proper settings and methods for the data analysis, and thus we are confident in our current data. Previous images have been replaced by higher quality versions of the same images, preserving 16-bit acquisition quality instead of 8-bit (Figures 3, 4, and 4—supplement 1) to address this point to the best of our ability.

Can engineering tapasin binding site/s into B35.1 rescue ER assembly of stable B35.1? In this regard, it is curious nothing is said or tested about the role of TAPBPR-a tapasin homologue that functions outside of the PLC in MHC-I/HLA-I. assembly of stable complexes.

This is an interesting suggestion. Tapasin-independent allotypes typically don’t differ in their tapasin binding site residues; rather the differences are largely conformational. Recent findings indicate that a closely-related pair of tapasin-dependent and tapasin-independent allotypes that have identical tapasin-binding sites (B*44:02 and B*44:05) are both able to bind tapasin, although with approximately 5-fold different affinities (6). We have previously measured increased stability (reduced aggregation) of peptide-deficient forms of tapasin-independent allotypes compared with tapasin-dependent allotypes (7), suggesting that the increased stability of peptide-deficient conformations of tapasin-independent allotypes could allow faster dissociation of suboptimal versions from the PLC. TAPBPR binding appears more focused on HLA-A compared with HLA-B (8). Very little has been done to examine the role of TAPBPR in endosomal assembly, and we believe this is an important topic of future study. However, this is beyond the scope of the current study, particularly based on the findings of new Figure 1, where we show that bafilomycin sensitivity of HLA-B in moDCs is not fully explained by TAP or tapasin dependencies.

MG132 is a reasonable proteasome inhibitor but it has limited specificity. Critical experiments should be repeated with lactacystin, which is a gold standard for proteasome inhibition; an alternative is epoxomicin, an irreversible proteasome inhibitor.It is quite evident that the two recombinant EBV-derived proteins are cross presented to respective T cells. The evidence in this report suggests that the cytoplasmic pathway dominates the endo-lysosomal pathway in cross-presentation. Using a broader spectrum proteasome inhibitor may help amplify the role of proteasomes and, hence, the cytoplasmic pathway to cross-presentation.

We thank the reviewer for these suggestions, based on which we have done additional cross-presentation experiments including lactacystin. These can be found in revised Figure 7D and 7E. These experiments revealed some epitopes are cross-presented in a lactacystin-sensitive manner while their presentation was MG132-resistant (suggesting possibly greater potency of lactacystin). The addition of this inhibitor further clarified that most epitopes in both monocytes and moDCs are presented via a cytosolic, proteasome-dependent pathway, with additional cathepsin-dependence of B*35:01-restricted epitopes. Thus, we agree that the cytosolic pathway dominates, and are grateful for the suggestion to clarify this. However, we also emphasize the combined sensitivity of B*35:01 to both proteasomal and cathepsin inhibition. We believe this result reflects the ability of this allotype to assemble in multiple compartments with exogenous antigens, thus providing greater flexibility for maintaining antigen presentation when the canonical ER pathway is inhibited. These points are described in in the Results section related to Figure 7 and the Discussion section (page 18).

Now, there is some evidence that autophagy can dump proteasomes into lysosomes. Considering that cross-presenting DCs are constitutively autophagic, which have roles in antigen cross-presentation by mouse MHC-I, are the outcomes of MG132 inhibition a result of proteasome activity in these late vesicles?

This is an interesting suggestion, which requires further study, and we cannot rule out this possibility based on our work. We suggest that it is unlikely that the outcomes of MG132/lactacystin inhibition in our experiments result from proteasome activity in lysosomes because the assembly status of B*08:01 prevents efficient peptide exchange in endosomes (demonstrated by a lack of peptide receptivity in Figure 2). We observe the bafilomycin-resistance of B*08:01 for its surface expression and peptide receptivity, indicating it does not utilize endo-lysosomal compartments for endogenous peptide exchange. Additionally, if B*08:01 were more capable of peptide exchange within endosomes/lysosomes during cross-presentation, we would also expect greater sensitivity to cathepsin inhibition, similarly to B*35:01.

Lastly, this work is narrowly focused on a single member of the B7 supertype-B*35:01, which is prevalent at low frequency in the African American (AA: 0.05) and White American (WA: 0.07) populations. Extension of current work beyond B35.1 and the Bw6 serotype by more quantitative cellular biochemical approaches to one or more additional allotypes in the B7 superfamily-e.g., B7.2 (AA: 0.08; WA: 0.155) itself and/or B35.3 (AA: 0.005; WA: 0.027) or B53.1 (AA: 0.133; WA: 0.004), could bolster and reveal to the importance of the findings in this report. Outcomes may also yield insights into the causes and maintenance of HLA-I polymorphisms.

We thank the reviewer for this suggestion, and agree with its importance. We have performed additional bafilomycin treatment studies shown in a new Figure 1, including six additional common HLA-B allotypes, following the recruitment of multiple donors for each allotype. The bafilomycin treatment revealed a spectrum of sensitivities to endo-lysosomal pH perturbation, although the sensitivity does not correspond strictly to tapasin-dependence or TAP-dependence as previously hypothesized. We suggest in the Results section corresponding to Figure 1 that there are several factors, including PLC-dependence and peptidome diversity, that could contribute to the individual abilities of allotypes to assemble with peptides in endo-lysosomal compartments.

Reviewer #3 (Recommendations for the authors):1) The introduction of the manuscript is rather long (5 paragraphs, 3.5 pages) for a research article. While this provides important details, it may also detract from the presentation of the results. Consider shortening and consolidating paragraphs in the introduction.

We thank the reviewer for this suggestion. The introduction has been considerably shortened to 3 paragraphs, 2 pages (specific changes are not highlighted). Please see Pages 3 and 4 for the new version. This suggestion has helped tighten the background.

2) P8, pp 2: Regarding the very elegant assay using HC10 can probe empty conformers: What are the background levels arising from staining of peptide-loaded molecules using this antibody for the two B allotypes, under the conditions of the experiments? One way to account for this is to incubate with either Bw6 or W6/32 first as blocking antibodies, and then measure background levels with HC10.

This is an important point, which we have addressed with a bead-based assay in new Figure 2D. Based on this comment, we wanted to ensure that recognition by the HC10 antibody was similarly reduced for both B*08:01 and B*35:01 upon addition of specific peptide. We conjugated purified biotinylated B*08:01 or B*35:01 bound to low affinity peptides to streptavidin beads, followed by pulsing with medium or high-affinity peptides or no peptides at various pH values to mimic different cellular conditions of peptide exchange. These data demonstrate that HC10 can recognize peptide-deficient conformers of both B*08:01 and B*35:01, and that specific peptides reduce empty conformer recognition by HC10 to equivalent levels, and based on peptide affinity for HLA-. Thus, the differences observed in Figures 2 and 4 must be due to factors in moDCs, such as the efficiency of ER assembly and endosomal recycling for each allotype. Based on the findings that HC10 signal changes are sensitive to peptide affinity, we also repeated the moDC B*08:01 HC10 peptide receptivity assays with the high affinity FLR peptide (Figures 2E and 4G), which again confirmed low peptide receptivity for B*08:01.

3) P11, pp3 and Figure 3G: HC10 suggests a reduction in B35 (but not B08) peptide receptivity upon bafilomycin treatment, measured by HC10 staining ratio. Are the receptive molecules properly conformed for peptide binding, and do they contain b2m? This is an important detail since the loss of b2m will leads to the formation of heavy chain dimers and re-internalization of MHC-I molecules (see, for example, Dirscherl et al., J cell Sci, 2022).

We thank the reviewer for this suggestion. The HC10 antibody-based peptide receptivity assays suggest an increase in B*35:01 (but not B*08:01) peptide receptivity (greater reduction in HC10 signal following the addition of specific peptide) in bafilomycin-treated cells. Since peptide binding requires β_2_m, we believe that the peptide receptive B*35:01 molecules are still bound to β_2_m. Indeed, in the HLA-B-beads-based assays, similar peptide conditions that induce a reduction in the HC10 signal (Figure 2D), also cause a parallel increase in the b_2_m-specific BBM.1 signal (Figure 4, supplement 1G and 1H).

This comment additionally prompted us to examine changes to surface β_2_m on moDCs upon bafilomycin treatment. New experiments (Figure 1H) show that after bafilomycin treatment, total surface β_2_m, as measured by the BBM.1 antibody, is reduced on moDCs in multiple donors. Thus, bafilomycin treatment appears to have a general negative consequence for the assembly and recycling of classical and/or non-classical MHC-I complexes, whereas some HLA-B allotypes such as B*08:01 and B*44:02 are resistant to the effects of bafilomycin. In repeating the peptide receptivity experiments using BBM.1, we found that B*35:01 peptides were unable to increase the surface β_2_m both in the presence and in the absence of bafilomycin (new Figure 4—supplement 1H and 1I). The detection of B*35:01 peptide receptivity by HC10 (Figures 2E and 4G) but not BBM.1 (new Figure 4—supplement 1H and 1I) is likely due to broader specificity of BBM.1 (which is β_2_m specific (9) and able to recognize classical and/or non-classical MHC-I complexes) compared with the HC10 antibody (which is specific for HLA-B and HLA-C (10)). We expect that the small peptide-induced changes in BBM.1 signal are more difficult to detect against the larger reduction in the total BBM.1 signal.

4) P13, pp1: the authors employ assays using (likely oligoclonal) tetramer-sorted T cells to probe the formation of properly conformed peptide:MHC-I complexes on the cell surface, however, there could exist cross-reactivity with different antigens which may convolute the results of this analysis. Please provide evidence for the lack of cross-reactivity for the different EBV epitopes arising from the same protein, using tetramer staining or a similar assay to complement the analysis.

We thank the reviewer for this comment. Peptide activation assays with each CTL were performed and no cross-reactivity between B*08:01 CTLs and B*35:01 peptide, or B*35:01 CTLs and B*08:01 peptide was detected. Results are shown in a new Figure 6—supplement 1.

5) P 14, pp3: the authors provide evidence of different response sensitivities for different EBV epitopes, depending on contributions from the cytosolic or vacuolar pathways. Can they elaborate further on plausible mechanisms for the observed differences, for example, changes in surface expression levels, densities, or distribution of molecules, and if possible, provide a quantitative assessment of the number of pMHC molecules under different conditions?

This is an important point that is further clarified in the relevant Results section (Pages 14-16). For a given HLA-class I allotype, the epitopes with lower response sensitivities have relatively lower affinities for HLA-class I (B*08:01-RAK, IC_50_ 366 nM and B*35:01-EPL, IC_50_ 534 nM) compared with the epitopes with the higher response sensitivities, which had higher affinities (B*08:01-FLR, IC_50_ 7 nM and B*35:01-YPL; IC_50_ 19 nM). This is likely due to lower cell surface levels achieved of the lower affinity complexes. Notably, consistent with the greater peptide receptivity for B*35:01 (Figure 2E), the T cell response sensitivity achieved with the lower affinity B*35:01-EPL epitope was similar to that achieved with the high affinity B*08:01-FLR epitope (Figure 6A). Furthermore, the data of Figure 6 and 7A-C show that with intact protein-derived epitopes as with the peptides, B*35:01 peptide loading during cross-presentation is more efficient than for B*08:01. Based on the additional analyses of Figure 7, we interpret these differences as attributable to the greater peptide receptivity of B*35:01 and its increased loading permissiveness in multiple cellular compartments.

A direct quantitation of pHLA molecules on the cell surface after cross-presentation by use of mass spectrometry and isotype-labeled standards would certainly emphasize this point, but that type of quantitation also has its limitations. For example, variations in peptide-HLA-class I dissociation rate or cell surface HLA-class I stability could influence the results, or some peptides may not be detectable by MS/MS of cell surface proteins. The extensive independent development that would be needed for this assay to be fully interpretable makes this experiment outside the scope of the present study.

6) Are there differences in glycosylation of MHC molecules at the conserved Asn 86 position processed through the vacuolar or cytosolic pathways? If so, it would be interesting to see if the contribution of different pathways, such as in the case of B08 vs B35 affects the complex glycosylation patterns on the cell surface.

This is an interesting point by the reviewer, and we agree that an examination of glycosylation and endosomal HLA-B trafficking would be beneficial. A previous study by our group (11) found that B*35:01 traffics to the surface of TAP-deficient cells in a Golgi-independent manner, as evidenced by immature glycans on B*35:01. We compared endo H sensitivities of cell surface HLA-class I in moDC’s from B*35:01 and B*08:01 donors and did not measure differences, and furthermore, we found that the majority of the cell surface HLA-class I (detected by HC10) is endo H resistant. These were new experiments we undertook for the revision in response to the comment, but we do not show these data as they are not directly relevant to the main points of the paper.

7) Figure 7 (the summary figure) is too complex, with too many graphical elements. For a conceptual model figure, this may confuse readers. Consider simplifying by reducing the number of pMHC complexes illustrated.

We thank the reviewer for this suggestion. The model figure (now Figure 8) has been greatly simplified to focus on B*35:01 recycling and peptide exchange in moDCs. This has allowed us to clarify the key takeaways from our study. We appreciate this comment.

References:

1. Sesma L, Galocha B, Vazquez M, Purcell AW, Marcilla M, McCluskey J, Lopez de Castro JA. 2005. Qualitative and quantitative differences in peptides bound to HLA-B27 in the presence of mouse versus human tapasin define a role for tapasin as a size-dependent peptide editor. *J Immunol* 174: 7833-44

2. Sesma L, Alvarez I, Marcilla M, Paradela A, Lopez de Castro JA. 2003. Species-specific differences in proteasomal processing and tapasin-mediated loading influence peptide presentation by HLA-B27 in murine cells. *J Biol Chem* 278: 46461-72

3. Huang M, Zhang W, Guo J, Wei X, Phiwpan K, Zhang J, Zhou X. 2016. Improved Transgenic Mouse Model for Studying HLA Class I Antigen Presentation. *Sci Rep* 6: 33612

4. Wang Z, Hu XZ, Tatake RJ, Yang SY, Zeff RA, Ferrone S. 1993. Differential effect of human and mouse β 2-microglobulin on the induction and the antigenic profile of endogenous HLA-A and -B antigens synthesized by β 2-microglobulin gene-null FO-1 melanoma cells. *Cancer Res* 53: 4303-9

5. Yarzabek B, Zaitouna AJ, Olson E, Silva GN, Geng J, Geretz A, Thomas R, Krishnakumar S, Ramon DS, Raghavan M. 2018. Variations in HLA-B cell surface expression, half-life and extracellular antigen receptivity. *eLife* 7

6. Jiang J, Taylor DK, Kim EJ, Boyd LF, Ahmad J, Mage MG, Truong HV, Woodward CH, Sgourakis NG, Cresswell P, Margulies DH, Natarajan K. 2022. Structural mechanism of tapasin-mediated MHC-I peptide loading in antigen presentation. *Nat Commun* 13: 5470

7. Rizvi SM, Salam N, Geng J, Qi Y, Bream JH, Duggal P, Hussain SK, Martinson J, Wolinsky SM, Carrington M, Raghavan M. 2014. Distinct Assembly Profiles of HLA-B Molecules. *J Immunol* 192: 4967-76

8. Ilca FT, Drexhage LZ, Brewin G, Peacock S, Boyle LH. 2019. Distinct Polymorphisms in HLA Class I Molecules Govern Their Susceptibility to Peptide Editing by TAPBPR. *Cell Rep* 29: 1621-32 e3

9. Parham P, Androlewicz MJ, Holmes NJ, Rothenberg BE. 1983. Arginine 45 is a major part of the antigenic determinant of human β 2-microglobulin recognized by mouse monoclonal antibody BBM.1. *J Biol Chem* 258: 6179-86

10. Stam NJ, Vroom TM, Peters PJ, Pastoors EB, Ploegh HL. 1990. HLA-A- and HLA-B-specific monoclonal antibodies reactive with free heavy chains in western blots, in formalin-fixed, paraffin-embedded tissue sections and in cryo-immuno-electron microscopy. *Int Immunol* 2: 113-25

11. Geng J, Zaitouna AJ, Raghavan M. 2018. Selected HLA-B allotypes are resistant to inhibition or deficiency of the transporter associated with antigen processing (TAP). *PLoS Pathog* 14: e1007171